# Psychosocial factors associated with mental health and quality of life during the COVID-19 pandemic among low-income urban dwellers in Peninsular Malaysia

Wong Min Fui[1,2], Hazreen Abdul Majid[2,3], Rozmi Ismail[4], Tin Tin Su[5], Tan Maw Pin[6], Mas Ayu Said[7] *

1 Training Management Division, Ministry of Health, Putrajaya, Malaysia, 2 Centre for Population Health (CePH), Department of Social and Preventive Medicine, Faculty of Medicine, University of Malaya, Kuala Lumpur, Malaysia, 3 Department of Nutrition, Faculty of Public Health, Universitas Airlangga, Jawa Timur, Indonesia, 4 Psychology and Human Well-being Research Centre, Faculty of Social Sciences and Humanities, University Kebangsaan, Bangi, Malaysia, 5 South East Asia Community Observatory (SEACO) and Global Public Health, Jeffrey Cheah School of Medicine & Health Sciences, Monash University, Bandar Sunway, Subang Jaya, Malaysia, 6 Department of Medicine, Faculty of Medicine, University of Malaya, Kuala Lumpur, Malaysia, 7 Centre for Epidemiology and Evidence-Based Practice, Department of Social and Preventive Medicine, Faculty of Medicine, University of Malaya, Kuala Lumpur, Malaysia

* mas@ummc.edu.my

**Data Availability Statement:** Data cannot be shared publicly because of the vulnerability of the

## Abstract

### Background and aims

Mental well-being among low-income urban populations is arguably challenged more than any other population amid the COVID-19 pandemic. This study investigates factors associated with depression and anxiety symptoms and quality of life among Malaysia's multi-ethnic urban lower-income communities.

### Methods

This is a community-based house-to-house survey conducted from September to November 2020 at the Petaling district in Selangor, Malaysia. Five hundred and four households were identified using random sampling, and heads of eligible households were recruited. Inclusion criteria were age ≥18 years with a monthly household income ≤RM6960 (estimated $1600) without acute psychiatric illness. The PHQ-9, GAD-7 and EQ-5D were used for depression, anxiety, and quality of life, respectively. Multivariable logistic regression was performed for the final analysis.

### Results

A total of 432 (85.7%) respondents with a mean age of 43.1 years completed the survey. Mild to severe depression was detected in 29.6%, mild to severe anxiety in 14.7%, and problematic quality of life in 27.8% of respondents. Factors associated with mild to severe depression were younger age, chronic health conditions, past stressful events, lack of communication gadgets and lack of assets or commercial property. While respiratory diseases,

population and sensitivity of the subjects. Data are available from University Malaya Research Ethics Committee (UMREC)(contact via Tel No.: 03-79676289 / 6942 e-mail: umrec@um.edu.my), Professor Mohd Azlan Shah Zaidi (azlan@ukm.edu.my) and Professor Mas Ayu Said (mas@ummc.edu.my) for researchers who meet the criteria for access to confidential data.

**Funding:** Long Term Research Grant (LRGS), LRGS/1/2016/UKM/02/1/2 (LGRS MRUN/F1/01/2019), Associate Professor Mas Ayu Said Malaysia Women Graduate Association (MWGA), PV076-2021, Dr Wong Min Fui Post-graduated Scholarship Award from the Federation under the Ministry of Health, Malaysia, Dr Wong Min Fui

**Competing interests:** The authors have declared that no competing interests exist.

marital status, workplace issues, financial constraints, absence of investments, substance use and lack of rental income were associated with mild to severe anxiety. Attributing poverty to structural issues, help-seeking from professionals, and self-stigma were barriers, while resiliency facilitated good psychological health. Problematic quality of life was associated with depression, older age, unemployment, cash shortage, hypertension, diabetes, stressful life events and low health literacy.

## Conclusions

A high proportion of the sampled urban poor population reported mild to severe anxiety and depression symptoms. The psychosocial determinants should inform policymakers and shape future work within this underserved population.

## Introduction

Depression and anxiety remain two major diagnosable common mental disorders contributing to disability and morbidity worldwide, partially attributable to a lack of public health investment in this area [1]. The World Health Organisation reported 800,000 suicide cases globally in 2015, with 78% of suicides occurring in lower to middle-income countries (LMICs) [2]. Almost half a million Malaysians have significant depressive symptoms that are particularly prominent among individuals living in households within the bottom 40% (B40) income bracket [3]. The recent coronavirus disease (COVID-19) pandemic has brought mental health issues to the forefront, with a reported increase in suicide cases from 609 cases in 2019 to 631 cases in 2020 during the pandemic period in this country [4, 5]. However, since there was an increase in country's population, thus no increase in suicide rates per capita was observed.

Over the past 50 years, the Malaysian government has invested in extreme poverty eradication and economic growth through shared prosperity within its multi-ethnic population. The rapid urbanisation observed from 1960 to 2010 had [6], however, led to the unintended consequences of the transformation of social structures resulting in pockets of urban poor in the cities of Malaysia. This has escalated poverty in urban areas within Selangor, the wealthiest state in Peninsular Malaysia [7], threatening urban residents' mental well-being [8]. Studies from high-income countries (HICs) and low-middle-income countries (LMICs) suggest that urban residents were more likely to develop neurotic conditions than rural residents [9–11]. In contrast, studies from China and Germany showed that rural residents were prone to mental health issues [12, 13]. Local studies also revealed a consistently high prevalence of depression (12.3% to 24.2%) and anxiety (18% to 36.3%) among low-income urban residents, with variations arising from screening tools utilised and selected cut-off scores [14–16].

Low socioeconomic status is a major social determinant of health (SDH) [17], which profoundly affects the morbidity and mortality of the community [18]. Education, ethnic group or social class, income, and employment are typical indicators of socioeconomic status [19]. Community-based studies from LMICs have revealed the relationship between low socioeconomic status (SES) and common mental disorders [20–23]. Recent literature revealed that low socioeconomic status was linked to lower health literacy with higher stigma [24, 25] and lack of mental health help-seeking [26]. Additionally, low-income individuals' perceptions of the structural issues attributed to inefficient government or discrimination may subsequently influence their decision-making and mental health [27]. Since evidence for these psychosocial

factors remains scarce, further exploration is needed to target mental health promotion among low-income populations [26].

The recent COVID-19 pandemic has crippled the economy and heightened the pre-existing financial strain among low-income populations. Undoubtedly, the government's containment measures prevented the spread of the coronavirus, but a significant number of people suffered from financial loss due to job dismissal and pay cuts [28]. Unemployment and financial issues are important stressors that can lead to depression [29]. Even though evidence showed stringent government containment measures moderate depression by promoting trust and easing uncertainty [30], timely implementation of early screening and treatment of mental health at-risk individuals should be in place. A recent meta-analysis suggests that individuals with pre-existing mood disorders are at higher risk of COVID-19 hospitalisation and death [31].

In addition, a study conducted during the COVID-19 pandemic revealed that those who showed avoidance and lower religious coping had a higher risk of developing mild to moderate depressive symptoms [32]. A recent study from China showed that resilience scores were inversely associated with mental health symptoms among subjects with mild COVID-19 [33]. This has led to concerns that the COVID-19 pandemic may adversely affect the mental health of populations in LMICs [34] that lack the resources to address the increase in mental health needs of their population [34, 35].

The World Bank has estimated that between 71 to 100 million people are being pushed into poverty due to the COVID-19 pandemic [35] and intensified inequality. Therefore, the current study seeks to measure the prevalence and psychosocial determinants of depression symptoms, anxiety symptoms and quality of life among the urban low-income population in Malaysia during the COVID-19 pandemic.

## Methods

### Setting

This was a community-based cross-sectional survey conducted from September 2020 until November 2020 (corresponds to the recovery movement control order (RMCO) and start of the Controlled Movement Control Order (CMCO)) (See Fig 1) at the Petaling district of the state of Selangor in Malaysia. Proportions and effect sizes were obtained from similar studies to estimate sample size (S1 Table). The sample size was estimated using Open Epi software with a significance level of 0.05 and a statistical power of 0.8.

A simple random sampling was done based on the household list provided by the research committee from the Department of Statistics Malaysia (DOSM). All respondents from the selected household a) aged 18 years and older with, (b) a household income of RM 6960 and below, and c) without any acute psychiatric illness were included in this study. Non- Malaysians were excluded from the study.

This study received ethical approval from the University of Malaya Research Ethics Committee (UMREC Non-Medical ref: UM.TNC2/UMREC– 811). Written consent was obtained from each eligible respondent prior to the enrollment.

### Data collection

Since this was a face-to-face data collection, each enumerator was briefed about the University of Malaya COVID-19 Fieldwork Safety Protocol. Trained enumerators administered validated questionnaires during the house-to-house data collection (see Fig 2).

The respondents completed multiple standardised instruments in the Malay language with assistance from the research team. The survey components comprised the socio-demographic domain, which captured the respondents' information such as age, gender, ethnicity,

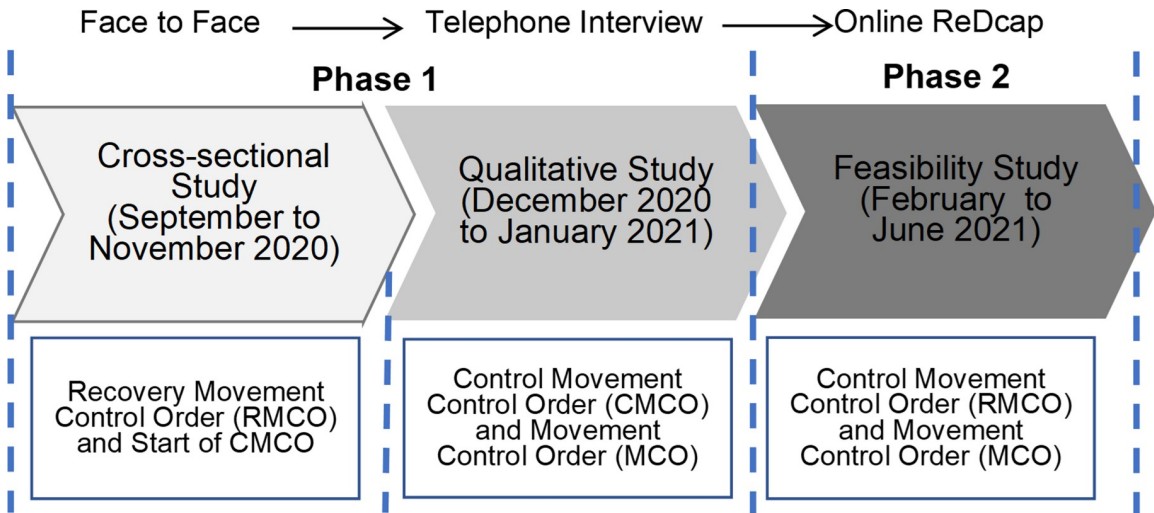

**Fig 1. Timeline of each phase of the study and the corresponding movement control.**

education level, employment status during the COVID-19 pandemic, monthly household income during the COVID-19 pandemic, marital status and household size.

The health domain components comprised self-reported weight and height, history of chronic illnesses, stressful life events, substance use, the 6-item Health Literacy Scale (HL-6) [36], 9-item Patient Health Questionnaire (PHQ-9) [37], 7-item Generalised Anxiety Disorders (GAD-7) [38] and the EQ-5D-5L health-related quality of life [39].

The presence of chronic diseases was self-reported by the participants based on ongoing medical attention or limiting daily activities, or both. The listed medical conditions included hypertension, diabetes mellitus, heart diseases, stroke, mental illness, and cancer. Substance use was recorded with a checklist of the top 10 most common substances, including tobacco, alcohol, cannabis, cocaine, amphetamine-type stimulant, inhalant, sleeping pills, hallucinogens, opioids, and other substances. The 13-item Malay version of the stressful life events checklist was also utilised [40].

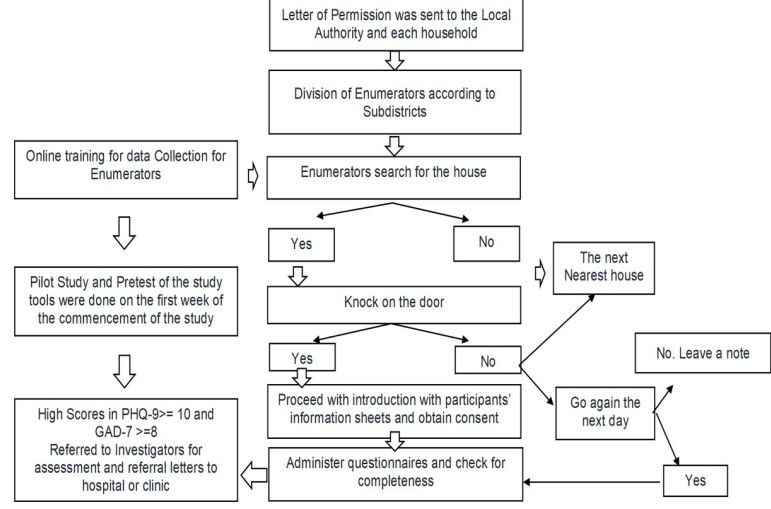

**Fig 2. The flow chart of the data collection.**

EQ-5D-5L is a standardised self-reported perceived health-related quality of life (QoL) which is rated through a descriptive system (EQ-5D) and a visual analogue scale (EQ-VAS), which captures perceptions on health status in the mobility, self-care, usual activities, pain or discomfort, and anxiety or depression domain. Each dimension is rated based on five response levels from no problems to unable to/extreme problems. Each health state is assigned a summary index score derived from a country-specific value set [41]. Health state index scores ranged from less than 0 to 1 (the value of full health), with higher scores indicating higher health utility. The visual analogue scale (EQ-VAS) records the respondent's overall current health on a vertical visual analogue scale on a scale of 0 to 100. The Malay language tool was validated by Shafie et al. (2018) with a fair agreement for convergent validity, while the English version has a Cronbach's alpha of 0.85 [39, 42].

The Patient Health Questionnaire measures depressive symptoms over the past two weeks. The PHQ-9 consists of 9 items scored across a four-point scale with a maximum score of 27. It is then classified further according to minimal (0–4), mild (5–9), moderate (10–14), severe (15–19), and very severe (20 or greater). Both English and Malay versions of the questionnaire have good internal reliability with a Cronbach's alpha of 0.70 [37, 43]. The total scoring methods were adopted [44].

Generalised Anxiety Scale also measures anxiety symptoms for the past two weeks. This was a 7-item instrument rated on a four-point scale. These cut-off values of minimal (0–4), mild (5–9), moderate (10–15) and severe (16–21) [45] were utilised. The Malay and English versions have good internal reliability, with a Cronbach's alpha ranging from 0.74 to 0.92 [38, 46].

Religiosity was determined with the Santa Clara Strength of Religious Faith Questionnaire 5 items (SCSRFQ-5) brief version, which was self-reported and rated on a 4-point Likert scale with a maximal score of 20 and a higher score indicating greater strength religiosity. The cut-off points for high ($\geq$ 17) or low (<17) religiosity were based on the sample median [47]. The Malay language of SCSRF has a Cronbach's alpha of 0.84 and was validated in the current research's pilot study with good internal reliability of Cronbach's alpha of 0.79 [48].

Resilience Scale of 14 items (RS-14) was derived from the 25 items scale developed by Wagnild et al. [49]. Each item was rated on a 7-point scale with a maximum total score of 98. The Malay version of the RS-14 has an excellent internal consistency of 0.86 [50].

Mental Help-Seeking Attitudes Scale (MHSAS) is a 9-item semantic scale with a higher score indicating a more positive attitude toward seeking help [51]. The Malay MHSAS has a Cronbach's alpha of 0.892 [52].

The Self-Stigma of Seeking Help (SSOSH) scale was developed by Vogel et al. (2006) and has 10 items rated on a 5-point scale [53]. The Malay SSOSH has been validated among the low-income group in Malaysia with an acceptable internal consistency of Cronbach's alpha = 0.667 [26] and also in the current study among the adult population (Cronbach's alpha = 0.84).

Poverty Attribution 21-items (PA-21) was used to measure the perception of low-income respondents on the cause of poverty. This 21-item new tool was developed by Ismail et al. (2019) and comprised structural support, socioeconomic support, individualistic and fatalistic domains [54]. Each item is rated on a 5-point scale, with higher scores indicating higher levels of agreement on the cause of impoverished conditions. The Malay language version showed acceptable internal validity of Cronbach alpha 0.61 to 0.87 among the four main domains [54].

Health Literacy Survey was a valid and reliable (Cronbach's alpha = 0.85) tool in six Asian countries [55]. Six items were extracted from the 12-item short-form version, with each item rated on a 4-point Likert scale. This tool was validated in the pilot study with good internal consistency (Cronbach's alpha = 0.898) and test-retest of inter-item covariance of 0.469 with

excellent Pearson correlation of r (98) = .839, p< .0001 were obtained. The total scores were computed and utilised for data analysis.

All study tools were validated in similar target populations but in different locations three weeks before the actual data collection. The Cronbach's alpha of each mentioned scale was at or, in most cases, well above the 0.7 thresholds illustrating strong consistency and, therefore, strong scale reliability (refer to S2 Table).

## Statistical analysis

### Data analysis

All statistical analyses were performed using the IBM SPSS Statistics for Windows, version 25.0 (IBM Corp LP, Armonk, NY, USA). Total scores of $\geq 5$ for PHQ-9 and GAD-7 were applied to determine the presence of symptomatic depression (depressed = 1, not-depressed = 0) and positive symptomatic anxiety (anxious = 1, not-anxious = 0), respectively. As for EQ-5D, the variable was dichotomised into no problem with QoL = 0 and problem with QoL = 1, using the cut-off score of 1.00. The students' independent sample T-test (normally distributed variables) and Mann-Whitney U-test (non-normally distributed variables) were conducted for univariate analysis. The Pearson's Chi-square or Fisher's Exact test was used to determine the strength of association between categorical explanatory variables and the outcomes variables. Multivariable logistic regression was performed to elicit significant final independent variables associated with PHQ-9, GAD-7 and EQ-5D. Hosmer-Lemeshow model development strategy was applied in the final analysis with variables selection criterion of p-values of less than 0.25 with backwards purposive variables exclusion of p-value more than 0.05 guided by the likelihood-ratio test [56]. Effect modification first-order main effects of explanatory variables were checked guided by the likelihood-ratio test. The finalised model was assessed for violation of assumptions of linearity of explanatory variables, log odds, multi-collinearity, and model fitness test. To ensure the effectiveness of the Hosmer Lemeshow test, the rule of thumb recommended by Paul et al. (2013) was applied, wherein a study with a sample size up to n = 1000, a number of groups up to 10 was used [57]. The model's sensitivity was also assessed using the area under receiving operating characteristics (AUROC) curve. Statistical significance was set at a p-value of less than 0.05.

## Results

### Participant characteristics

Of 504 eligible participants, 432 (82.7%) completed the survey. Table 1 summarises the participants' characteristics. Respondents had a mean age of 43.1 (SD 13.2) years. The unemployment rate doubled (33.2%) during the pandemic while income was reduced by 13.5%, and 41% lived below the poverty line.

Approximately 133 (31.2%) respondents had problematic literacy levels, and 29 (6.8%) had an inadequate level of health literacy. The professional help-seeking median score was high at 6.0 (IQR 2), and the mean self-stigma was neutral at 3.0 (SD 0.6). Poverty attribution of structural (3.6 (SD 0.8)) and socioeconomic(3.8 (SD 0.9)) domains gathered the highest mean scores out of the four domains of PA-21. Low resiliency was found in 133 (31%) participants. A total of 182 (42%) scored 17 and below for religiosity. Of the total respondents, 432 (30.4%) had a self-reported history of non-communicable diseases with hypertension in 78 (18.2%), diabetes in 38 (8.9%) and respiratory disease (COPD or Asthma) in 20 (4.7%). 172 (40.7%) were overweight (body mass index (BMI) >23.0–27.4 kg/m$^2$) and 29.6% obese (BMI $\geq$27.5kg/m$^2$). Among 130 (30.2%) respondents, substance use was present, with tobacco, alcohol, and

**Table 1. Socio-demographic profiles of low-income respondents from the Petaling district (n = 432).**

| Variables | n | % |
|---|---|---|
| **Gender** | | |
| Male | 278 | 64.5 |
| Female | 153 | 35.5 |
| **Age group (years old)** | | |
| Below 30 | 81 | 19.8 |
| 30–40 | 114 | 27.8 |
| 41–50 | 104 | 25.4 |
| More than 50 | 111 | 27.0 |
| **Ethnicity** | | |
| Malay | 323 | 75.1 |
| Chinese | 58 | 13.5 |
| Indian | 31 | 7.2 |
| Others | 18 | 4.2 |
| **Marital status** | | |
| Single | 75 | 17.5 |
| Married | 321 | 74.7 |
| Widowed | 34 | 7.8 |
| **Education level categorical** | | |
| No Formal & Primary School | 29 | 6.8 |
| Secondary Schools | 206 | 48.4 |
| More than Secondary Schools | 191 | 44.8 |
| **Are you currently employed or working (Before COVID-19)?** | | |
| No | 69 | 16.2 |
| Yes | 357 | 83.8 |
| **Are you currently employed or working (During COVID-19)?** | | |
| No | 137 | 33.2 |
| Yes | 276 | 66.8 |
| Household income (before Covid-19) | | |
| <RM 1700 | 86 | 19.9 |
| MYR 1700 to 2700 | 100 | 23.1 |
| MYR 2701 to 3700 | 97 | 22.5 |
| MYR 3701 to 4700 | 62 | 14.4 |
| MYR 4700–5700 | 36 | 8.3 |
| More than MYR 5701 | 51 | 11.8 |
| **Household income (during Covid-19)** | | |
| Below MYR 1700 | 136 | 31.5 |
| MYR 1700 to 2700 | 91 | 21.1 |
| MYR 2701 to 3700 | 77 | 17.8 |
| MYR 3701 to 4700 | 55 | 12.7 |
| MYR 4700–5700 | 32 | 7.4 |
| More than MYR 5701 | 41 | 9.5 |
| **Household size (person)** | | |
| Less than 4 | 191 | 47.8 |
| More than or same as 4 | 234 | 55.2 |
| **House-ownership** | | |
| Owners | 179 | 41.7 |
| Shelter | 13 | 3.0 |

(*Continued*)

**Table 1.** (Continued)

| Variables | n | % |
|---|---|---|
| Renting | 219 | 51.1 |
| Inherited | 18 | 4.2 |
| **Asset** | | |
| House | 231 | 53.8 |
| Vehicle | 350 | 81.8 |
| Land | 42 | 9.8 |
| Orchard | 23 | 5.4 |
| Cash | 148 | 34.5 |
| Rental house | 55 | 12.9 |
| Jewellery | 128 | 29.9 |
| Shophouse | 6 | 1.4 |
| Investment | 49 | 11.4 |
| Others | 4 | 0.9 |
| **Communication tools (own a mobile or smartphone)** | | |
| Yes | 340 | 81.9 |
| No | 75 | 18.1 |
| **Own at least a form of communication tools (TV or radio and internet or laptop and smartphone/mobile phone)** | | |
| Yes | 405 | 97.59 |
| No | 10 | 2.41 |

sleeping pills being the most used substances. Approximately 209 (48.5%) had a history of stressful life events, including losing loved ones, followed by working environment issues and job loss. Mild to severe depression symptoms were reported by 127 (29.6%), with 30 (7%) of whom had moderate to severe depressive symptoms. There were mild to severe anxiety symptoms in 63 (14.6%), with 19 (4.4%) reporting moderate to severe depressive symptoms. The descriptive findings for psychosocial risk factors and health-related profiles can be found in S3 and S4 Tables, respectively.

## Multivariable analyses

Factors positively associated with mild to severe symptoms of depression include being within the age group of "less than 30 years" (OR 5.11 (95%CI 2.04, 12.83)), self-reported hypertension, having other chronic illnesses, and having the presence of past stressful life events (physical assault, long term illness, family issues and workplace issues). Protective factors against the development of mild to severe depressive symptoms were those who owned one or more communication tools (television or radio and internet or laptop and smartphone/mobile), absence of assets such as investment shares, shop-houses, attributing structural issues to poverty, increase in resilience scores and those who finds professional help is beneficial. Factors associated with anxiety symptoms were respiratory illness (COPD or Asthma), stressful life events (marital, financial, and workplace issues), and sleeping pills. Perceived structural issues related to poverty, less self-stigma, and higher resilience scores were associated with fewer anxiety symptoms (Table 2).

Quality of Life (QoL) was problematic in 119 (27.8%). The EQ-5D has an overall median score of 1.00 (0.8) and a mean score of 0.94 (SD: 0.12). Out of the 3125 possible health states with the EQ-5D-5L, 54 health profiles were reported. Out of the 54 health profiles, 72.2% of patients reported a complete health state of 11111 ("no problem" with quality of life), followed

**Table 2. Simple and multivariable logistic regression on factors associated with depression (n = 385) and anxiety symptoms (n = 398) (cut-off at 5).**

| Factors | Depression (PHQ-9) | | Anxiety (GAD-7) | |
|---|---|---|---|---|
| | Crude OR (95% CI) | Adjusted OR (95% CI) | Crude OR (95% CI) | Adjusted OR (95% CI) |
| **Age** | | | | |
| Below 30 | Reference | Reference | Reference | Reference |
| 30–40 | 0.45 (0.24, 0.83)* | 0.52 (0.24, 1.12) | 0.68 (0.32, 1.45) | 0.78 (0.31, 1.92) |
| 41–50 | 0.46 (0.24, 0.88)* | 0.47 (0.21, 1.09) | 0.51 (0.22, 1.17) | 0.68 (0.25, 1.88) |
| More than 50 | 0.35 (0.18, 0.67)* | 0.20 (0.08, 0.49)* | 0.47 (0.20, 1.07) | 0.41(0.15, 1.12) |
| **Asset-Shop lot** | | | | |
| Yes | Reference | Reference | | |
| No | 0.08 (0.01, 0.70)* | 0.08 (0.07, 1.00) | | |
| **Asset-Investment** | | | | |
| Yes | Reference | Reference | | |
| No | 0.52 (0.28, 0.95)* | 0.37 (0.17, 0.80)* | | |
| **Asset-Rental house** | | | | |
| Yes | | | Reference | Reference |
| No | | | 0.65 (0.32, 1.34) | 0.44 (0.18, 1.06) |
| **Hypertension** | | | | |
| No | Reference | Reference | | |
| Yes | 1.57 (0.94, 2.63) | 2.25 (1.01, 5.00)* | | |
| **Other diseases** | | | | |
| No | Reference | Reference | | |
| Yes | 7.29 (2.78, 19.10)** | 4.06 (1.26, 13.06)* | | |
| **Asthma** | | | | |
| No | | | Reference | Reference |
| Yes | | | 2.12 (0.74, 6.07) | 4.44 (1.18, 16.67)* |
| **Sleep pill usage** | | | | |
| No | | | Reference | Reference |
| Yes | | | 10.46 (2.43, 44.95)* | 11.89 (1.26, 112.16)* |
| **Physical or sexual assault** | | | | |
| No | Reference | Reference | | |
| Yes | 5.68 (2.38, 13.54)** | 4.77 (1.56, 14.57)* | | |
| **Prolonged or serious illness** | | | | |
| No | Reference | Reference | | |
| Yes | 5.24 (2.45, 11.236)** | 4.892 (1.834, 13.053)* | | |
| **Family Issues** | | | | |
| No | Reference | Reference | | |
| Yes | 6.87 (2.94, 16.06)** | 3.74 (1.26, 11.14)* | | |
| **Marital Problem** | | | | |
| No | | | Reference | Reference |
| Yes | | | 8.40 (3.45, 20.44)** | 4.73 (1.52, 14.70)* |
| **Financial issues** | | | | |
| No | | | Reference | Reference |
| Yes | | | 8.45 (3.70, 19.32)** | 4.67 (1.46, 14.93)* |
| **Workplace Issue** | | | | |
| No | Reference | Reference | Reference | Reference |
| Yes | 3.28 (1.78, 6.04)** | 3.48 (1.58, 7.67)* | 5.03 (2.61, 9.71)** | 3.533 (1.53, 8.17)* |
| **Owing Communication Gadget 1 or 2** | | | | |
| No | Reference | Reference | | |

*(Continued)*

**Table 2.** (Continued)

| Factors | Depression (PHQ-9) | | Anxiety (GAD-7) | |
|---|---|---|---|---|
| | Crude OR (95% CI) | Adjusted OR (95% CI) | Crude OR (95% CI) | Adjusted OR (95% CI) |
| Yes | 0.27(0.07, 0.97)* | 0.06 (0.01, 0.38)* | | |
| **Poverty attribute-Structural** | 0.65 (0.51, 0.83)** | 0.66 (0.48, 0.91)* | 0.78 (0.58, 1.05) | 0.67 (0.45, 0.99)* |
| **MHSAS (Total)** | 0.96 (0.94, 0.98)** | 0.96 (0.94, 0.99)* | 0.98 (0.95, 1.00) | 0.99 (0.97, 1.03) |
| **SSOSH (Total)** | | | 1.12 (1.06, 1.19)** | 1.09 (1.01, 1.17)* |
| **Resilience (Total)** | 0.96 (0.94, 0.97)** | 0.96 (0.94, 0.98)** | 0.96 (0.94, 0.98)** | 0.97 (0.95, 0.998)* |

OR, Odds ratio; CI, Confidence Interval; PHQ-9, 9-item Patient Health Questionnaire; GAD-7, 7-item Generalised Anxiety Disorders; EQ-5D, European health-related quality of Life- Five Domains; MHSAS, Mental Help-Seeking Attitudes Scale; SSOSH, Self-Stigma of Seeking Help.

** $p = <0.001$

* $p = <0.05$.

PHQ-9: Nagelkerke's R square = 0.372; Hosmer and Lemeshow Test = 0.382; Area under the curve = 0.837 $p = <0.001$.

GAD-7: Nagelkerke's R square = 0.307; Hosmer-Lemeshow goodness of fit test = 0.888., Area under the curve = 0.799 (95% CI: 0.737, 0.861, $p = <0.001$).

by 4.91% reporting a "problem" health state of 11121 and 3.04% for 11122. Pain/ discomfort and depression/anxiety were two domains that captured the higher "problematic" frequencies compared to other domains.

Mild to severe depressive symptoms were associated with increased odds of problematic QoL. Age, female sex, unemployment, hypertension, diabetes, and stressful life events of severe injury were associated with poorer QoL. Higher literacy scores were protective against poorer QoL (Table 3). Refer to S5 Table (Tables A-I for the details of univariable analysis and Tables J-L for the multivariable variable selection process).

## Discussions

Comparatively, depression and anxiety levels were higher than those previously reported in the National Health and Morbidity Survey (NHMS) 2012–2019 [3, 58]. Compared to other local studies, anxiety levels were lower than the pandemic levels [28, 59, 60]. While mild to moderate anxiety levels were higher than those previously reported in international studies [61–63], they were also lower than those conducted in other countries during the pandemic [64, 65]. Lower-income groups with higher financial strain respondents were more likely to experience mild to severe depression [28]. However, lower anxiety levels could be attributed to differences in assessment tools [28, 59, 60]. As the study was conducted towards the later stages of the pandemic as opposed to previous studies [66, 67], anxiety levels could have been higher initially due to difficulties adapting to containment measures and fear of infection and death [68]. Depression may be more prominent at later stages when the future is uncertain due to job losses, pay cuts and poor social support [69, 70].

To date, only a few studies have examined the relationship between mental health issues and quality of life for low-income groups that allowed for comparison. The majority were those studies targeted at the general population with specific diseases [71]. The overall prevalence for poorer quality of life was 27.8%, comparable to a validation study from Trinidad and Tobago (28.0%) [72] but lower than most of the pre-pandemic studies from other countries (EQ-5D: 54.0%-69.7%) [73–76]. The figure remains low even after accounting for socio-demographic and chronic illness characteristics [73, 77]. Out of this figure, pain/discomfort and anxiety/depression domains had the most significant number of problematic QoL respondents.

**Table 3. Simple and multivariable logistic regression on factors associated with EQ-5D (n = 385).**

| Factors | Crude OR (95% CI) | Adjusted OR (95% CI) |
|---|---|---|
| **Gender** | | |
| Male | Reference | Reference |
| Female | 1.61 (1.05, 2.49)* | 2.34(1.35, 4.04)* |
| **Age group (year-old)** | | |
| Below 30 | Reference | Reference |
| 30–40 | 1.09 (0.51,2.34) | 1.21 (0.49, 2.99) |
| 41–50 | 1.53 (0.71, 3.28) | 1.61(0.63, 4.10) |
| More than 50 | 3.24 (1.56, 6.74)* | 3.06 (1.17, 8.03)* |
| **Asset-Cash** | | |
| Yes | Reference | Reference |
| No | 1.61(1.01, 2.56)* | 1.88 (1.04, 3.39)* |
| **Work during outbreak** | | |
| Yes (R) | Reference | Reference |
| No | 2.33(1.49, 3.64)** | 1.34 (0.76, 2.38) |
| **Hypertension** | | |
| No | Reference | Reference |
| Yes | 4.18 (2.49, 6.99)** | 2.56 (1.22, 5.26)* |
| **Diabetes** | | |
| No | Reference | Reference |
| Yes | 5.98 (2.93, 12.22)** | 3.07 (1.22, 7.71)* |
| **Severe injury due to accident** | | |
| No | Reference | Reference |
| Yes | 2.88 (1.41, 5.86)* | 3.59 (1.37, 9.42)* |
| **Jobless** | | |
| No | Reference | Reference |
| Yes | 2.47 (1.35, 4.59)* | 2.54 (1.20, 5.37)* |
| **Depression-symptomatic** | | |
| < 5 (R) | Reference | Reference |
| ≥ 5 | 3.31 (2.11, 5.18)** | 2.79 (1.57, 4.49)** |
| **Health literacy index** | 0.96 (0.94, 0.98)** | 0.96 (0.94, 0.99)* |

OR, Odds ratio; CI, Confidence Interval.

** $p = < 0.001$

* $p = <0.05$.

Nagelkerke's R square = 0.345; Hosmer and Lemeshow Test = 0.857; Area under the curve = 0.813 (95% CI: 0.765, 0.861), p = < 0.001.

The EQ-5D mean score of 0.94 from the current study is comparable to studies done by Tran et al. (2020) and Vu et al. (2020) among the general population in Vietnam during the COVID-19 pandemic [78, 79]. A higher proportion of participants had full health scoring (72.2%) compared to participants from Vietnam (54.9% to 60.0%) [80]. The mean score of EQ-5D was also found to be better than those studies which focused on chronic diseases in Malaysia and other countries, which entails patients who have diabetes [80], human immuno-deficiency virus (HIV) [81], skin diseases [82], respiratory diseases [83] dengue fever [84], frail elderly [85], elderly after fall injury [86] fracture injuries [87] and Chronic Myeloid Leukaemia [88]. The possible explanation for the lower figure is the higher proportion of Malay and male participants in this study sample. In a separate cross-tabulation analysis done in the present

study, anxiety/depression was the only domain that showed significant difference across ethnic groups and gender, with Malay and males inclined to report "no problem" to their mental health well-being. Another local validation study observed a similar trend [89]. Therefore, the better-perceived health-related quality of life among B40 for the current study is worth further exploration in future studies to rule out possible information bias due to cultural influence on the lack of disclosure of mental health issues.

A positive association between higher scores of depression and poorer quality of life was established in this study. These results were similar to studies conducted in Malaysia, China and Slovenia involving urban community samples [28, 90, 91]. Abdullah et al. (2021) from the northwest coast of Peninsular Malaysia conducted a similar study with different tools, and their results varied for depression and anxiety among the domains in WHOQoL-BREF [59]. Another large scale mental health study National Epidemiologic Survey on Alcohol and Related Conditions (NESARC) [92], has shown anxiety is not significantly associated with quality of life. Variation in methodology in terms of the measurement tools and location of the studies may explain the inconsistency of findings.

The escalation of financial burdens due to pay cuts and unemployment has led to marital and family issues during the pandemic, increasing depression and anxiety symptoms. This was supported by a Malaysian study done in urban settings that revealed severe problems at work, unhappy relationships with children, spouse and family, and severe financial constraints are stressful life events for depression [29]. The same investigator also measured anxiety symptoms and found that unhappy relationships with family and severe problems at work predicted the outcomes [93]. The odds ratio for stressful life events was two times higher than those pre-pandemic findings reported by Kader et al. (2014) [29]. No doubt, movement restriction increases contact time among the family members, but it can aggravate pre-existing family conflict and causes stress to the vulnerable population [94]. In line with the current study's findings, physical or sexual assault positively correlated with depression. Woman's Aid Organisation (WAO) and Women, Family and Community Development Ministry reported increased usage of their public hotlines during the pandemic, and domestic violence was the main reason for calling [95].

Additionally, Malaysia's divorce rate also rose from 60,088 cases in 2017 to 90,766 cases in 2020, based on the Syariah court data [96]. The majority of the marital issues involved cases from the bottom 40% of the low-income group who faced challenges of job loss and financial crisis [97]. Economic difficulty is closely related to depression among the parents [98]. Therefore, it is pertinent to look out for stressful life events like a history of child abuse or intimate partner physical abuse as a critical risk factor for mental health issues among the low-income group.

Younger respondents (less than 30 years old) have a higher risk of developing depression symptoms than older respondents during the pandemic. The social, emotional and cognitive maturation were observed by neurodevelopmental scientists even right before the adolescent age extending to the 20 to 30 years of age [99, 100]; this has marked the vulnerability of the brain towards environmental changes and insults during this transition period towards young adulthood will tip-off mental health issues. Three studies were conducted in the local community, and the general population in urban settings supported the relationship between the younger age group of respondents and a higher likelihood of mental health issues [14, 15, 28]. Likewise, similar findings are notable among studies in other countries [101–103]. Several community-based studies have revealed a high prevalence of mental health problems among young adults, especially among students [104, 105]. Prolonged school closures and a switch to online learning occurred because of lockdown and social distancing measures. Those within the local socioeconomic classes were disadvantaged as a result of limited access to good internet connections and electronic devices [105].

Gender did not reveal any positive findings for mental health status in this study. These were unexpected findings as females were the most replicable risk factors in past studies among the general population [3, 106, 107], low-income groups [21, 108], and during the pandemic [101, 109]. Perhaps the COVID-19 pandemic may have put low-income families under a serious financial strain, leading to emotional turmoil for both genders. On the other hand, the older age group (more than 50 years) and females had a poorer quality of life. The plausible explanation is the higher probability of getting chronic illness at an older age, and chronic illness tends to be higher among females than males based on the NHMS 2019 data [3].

The distribution of the top two commonest non-communicable diseases, hypertension and diabetes, in this study is comparable to those found among the general urban population reported in this country's population-based survey NHMS 2019 [3]. The prevalence of depression in this study is comparatively higher than in the NHMS 2019 but lower than in similar studies in the LMICs [9, 110]. The higher rate observed in the latter study was attributed to hospital-based samples and underdeveloped mental health services for secondary prevention in lower-income countries.

Respondents with a known history of hypertension, other diseases and perceived chronic illness as stressful life events were likely to report higher depressive symptoms. Asthma was associated with a higher risk for anxiety symptoms [111, 112]. However, out of all the self-reported illnesses, only hypertension and diabetes were associated with poorer QoL. Studies showed that physical illnesses like cardiovascular-related diseases [113, 114], obesity and metabolic syndrome [115] had been proven to predict common mental health issues and affect the quality of life among the low-income population. The mechanisms underlying the causal relationship between mental and physical health are multifactorial that entail biological, psychosocial, environmental and behavioural. Environmental factors that induce chronic stress (psychosocial risk factors) may promote physical illness such as obesity due to unhealthy eating and a sedentary lifestyle. Through the psychosocial pathway, the physiological feedback from environmental stress factors results in the production of intermediate markers like pro-inflammatory markers interleukin (IL)–6 and tumor necrosis factor-alpha (TNF–$\alpha$) [116], vascular stiffening and endothelial dysfunction leading to adverse cardiovascular outcomes [117]. In obesity, high visceral fat is the major site for deposition of (IL)-6, which explain the association between depression, inflammation, metabolic risk factors and cardiovascular diseases [111, 118]. Therefore, the long-standing financial strain concurrent with the non-communicable disease may contribute to the mental health problems experienced by the B40 community and was explained by the psychosocial risk factors model.

Respondents reported a high proportion of low health literacy, with positive findings and poorer life quality. This evidence supported the high rate of chronic illness among the low-income group revealed by publications from PARTNER's study [119–122]. However, collective evidence from the meta-analysis showed heterogeneous results for the association between health literacy and quality of life among developed and developing countries [123]. The authors attribute the outcomes to variation in the health literacy tool among the studies; therefore, more studies are needed to provide robust evidence to support the relationship. Given the high prevalence of non-communicable diseases and low health literacy in this population, timely, reliable health information is critical during pandemic for early detection by recognising symptoms of ill health conditions or warning signs of COVID-19 and reducing fear. Strong partnership with the key community leaders is important to disseminate reliable information which is adapted to the local languages in simple presentation and accessible to the low-literacy community [124].

Based on the body of evidence, stigma is one of the well-established determinants for help-seeking barriers among the population from low and high-income countries. The majority of

the respondents from this study presented with a neutral score for self-stigma. The final analysis revealed, that those with higher self-stigma scores were likely to have anxiety symptoms, whereas lower mental health-seeking attitudes predicted higher depressive symptoms. Poor help-seeking attitude prevents the low-income respondents from getting earlier treatment for severe mental health issues. The negative evaluations of professionals derived from negative past experiences and mistrust of the mental health professionals possibly deter a person from seeking help [125]. Data from the Institute's mental health services policy analysis showed an overall limited mental health workforce in this country [126]. Currently, no permanent counsellors are available to handle mild to moderate mental health cases at the primary care clinic. Existing primary healthcare providers who were not trained in mental health were obligated to manage such cases who were first diagnosed with mental health issues. This may spill over public trust in professional help-seeking whenever a mismanaged case occurs.

The Let's Talk campaign was launched recently as part of health promotion from Malaysia's Ministry of Health, which aimed to destigmatise mental illness through public education and encourage people to seek help. However, more well-defined areas of focus and evidence-based strategies are still needed. For instance, Japan and Hong Kong have standardised and advocated the use of less stigmatised terms, while China has enacted national legislation and public education emphasizing the need to respect psychiatric patients as an anti-discriminatory approach [127]. To date, internet-based cognitive behavioural therapy (CBT) is the most documented evidence-based, effective in psychiatric symptoms alleviation [128] and cost-effective [129] psychological intervention, which has been embraced into the mainstream to address the mental health burden during the pandemic [130].

The measures of perception of causes of poverty (mean scores of 3.6) revealed the respondents' agreement with the structural barriers, but those who refuted poverty as a structural issue showed a reciprocal association for depressive and anxiety symptoms. Based on limited data and small sample size studies, poor respondents are more likely to endorse structural or external attribution for poverty [131, 132], and those who endorsed structural issues are more likely experiencing mental health problems [27], in contrast to the current findings. Ismail et al. (2019) postulated that respondents who do not blame the government are more self-sufficient and thus experience better financial well-being [54]. Pandemic may play a crucial role in the inconsistency of the findings, and the available evidence is not robust for a sound conclusion. Therefore, further study is required to explore the low-income individuals' perception of their motivation to overcome poverty [27]. Short-term stipends from the government such as Bantuan Prihatin Rakyat (BPR) in the form of fast cash may ease their financial stress in the short term, but life skill training and sustainable income-generating work are needed to get them out of the poverty cycle. While allowing workers to return to work during a pandemic eased the financial strain, this conversely increased infection risk. Studies suggest that implementing psychoneuroimmunological preventive measures at the workplace, which entails practising hand hygiene, maintaining social distancing, wearing a facemask and a healthy lifestyle, may ease the psychological distress and smoothen the process of return to work [133, 134].

Approximately one-third of respondents had low resilience. Higher resilience scores were associated with lower depressive and anxiety symptoms. The findings were supported by a meta-analysis revealing of the association between inversed resilience scores with depression and anxiety [135]. A 4 years longitudinal study involving a sample of 314 college students in China supported a reciprocal relationship between resilience and mental ill-being [136]. Respondents from this study were vulnerable, given their low-income status, and almost half of them reported a history of stressful life events. Building resiliency through health promotion is crucial to shield them from the impact of ill mental well-being due to adversity at the verge of the current pandemic financial crisis.

This is one of the first studies comprehensively exploring the various psychological risk factors during the COVID-19 pandemic among the low-income group in LMICs. This is a face-to-face study, thus having the benefits of capturing the responses of the non-technology savvy subjects compared to many other online studies done during the pandemic. It also improves rapport and eases the process of obtaining consent. As this was a cross-sectional study, the temporal causal relationships between the independent variables to depression, anxiety and quality of life could not be assigned. The majority of the sample were heads of household, male and older age group, the findings may not be generalisable to all lower-income populations. In addition, most of the studied factors were not related to COVID-19 hence the future research direction should focus on the effect of physical symptoms [137], facemask use [138], discrimination related to COVID-19 positive cases [139], higher numbers of children in the family [140], cross-cultural belief or religiosity [141], and impact of excessive exposure of the COVID-19 related health information on mental health [142].

## Conclusions

Mental health problems compromise the quality of life of the low-income group. The prevalence of depression and anxiety symptoms was higher than in studies conducted prior to the pandemic. Apart from socio-demographic factors, chronic illnesses and stressful life events, this study unveiled psychological barriers and facilitators such as stigma, help-seeking behaviour and resiliency for mental health. These outputs provide suitable targets for subsequent psychosocial intervention development within low-income communities. This is much needed to improve mental health status and empower the low-income population, ensuring that they are able to thrive during this challenging pandemic period and beyond.

## Supporting information

**S1 Table. Sample size according to study objectives.**
(PDF)

**S2 Table. Reliability of the study tools.**
(PDF)

**S3 Table. Psychosocial risk factors of the B40 respondents from the Petaling district.**
(PDF)

**S4 Table. Health-related profiles of B40 respondents from the Petaling district.**
(PDF)

**S5 Table. Univariable analysis results tables and multivariable variable selection process.**
(PDF)

## Acknowledgments

We would like to thank the Director General of Health Malaysia for his permission to publish this article.

We are grateful to Ng Yit Han and Nithiah Thangiah for their contribution in statistical analysis.

## Author Contributions

**Conceptualization:** Wong Min Fui, Hazreen Abdul Majid, Rozmi Ismail, Tin Tin Su, Tan Maw Pin, Mas Ayu Said.

**Data curation:** Wong Min Fui.

**Formal analysis:** Wong Min Fui.

**Funding acquisition:** Hazreen Abdul Majid, Tin Tin Su, Tan Maw Pin, Mas Ayu Said.

**Investigation:** Wong Min Fui.

**Methodology:** Wong Min Fui, Hazreen Abdul Majid, Rozmi Ismail, Tin Tin Su, Tan Maw Pin, Mas Ayu Said.

**Project administration:** Wong Min Fui, Hazreen Abdul Majid, Tin Tin Su, Tan Maw Pin, Mas Ayu Said.

**Resources:** Wong Min Fui, Hazreen Abdul Majid, Rozmi Ismail, Tin Tin Su, Mas Ayu Said.

**Supervision:** Hazreen Abdul Majid, Rozmi Ismail, Mas Ayu Said.

**Validation:** Wong Min Fui, Hazreen Abdul Majid, Rozmi Ismail, Mas Ayu Said.

**Writing – original draft:** Wong Min Fui.

**Writing – review & editing:** Hazreen Abdul Majid, Rozmi Ismail, Tin Tin Su, Tan Maw Pin, Mas Ayu Said.

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
