## [Decision Letter · Decision Letter 0]

4 May 2022

PONE-D-22-05096Psychosocial factors associated with mental health and quality of life during the COVID-19 pandemic among low-income urban dwellers in Peninsular MalaysiaPLOS ONE

Dear Dr. Said,

Thank you for submitting your manuscript to PLOS ONE. After careful consideration, we feel that it has merit but does not fully meet PLOS ONE’s publication criteria as it currently stands. Therefore, we invite you to submit a revised version of the manuscript that addresses the points raised during the review process.

We look forward to receiving your revised manuscript.

Kind regards,

Arsham Alamian, PhD, MSc, FACE, FRSPH

Academic Editor

PLOS ONE

Journal Requirements:

Reviewers' comments:

Reviewer's Responses to Questions

**Comments to the Author**

1. Is the manuscript technically sound, and do the data support the conclusions?

Reviewer #1: Yes

Reviewer #2: Yes

2. Has the statistical analysis been performed appropriately and rigorously? 

Reviewer #1: Yes

Reviewer #2: Yes

3. Have the authors made all data underlying the findings in their manuscript fully available?

Reviewer #1: Yes

Reviewer #2: Yes

4. Is the manuscript presented in an intelligible fashion and written in standard English?

Reviewer #1: Yes

Reviewer #2: Yes

5. Review Comments to the Author

Reviewer #1: Abstract

- No comments or suggestions

Introduction

- Line 58: replace “suicides” with “suicide”

- Line 60: replace “Malaysian” with “Malaysians”

- Line 76: remove “on”

- Line 91-93: Rephrase as it does not make sense

- Line 99: add “pandemic” after “COVID-19”

Methods

- Manuscript states that the study was conducted from September to November of 2020 in the introduction section but

from September to November of 2021 in the methods section. When did this study take place? Clarify and ensure that

the information is identical across the manuscript.

- Alpha value of 0.2 is relatively high resulting in an increased probability of making a type I error

- Line 117: replace “enrolment” with “enrollment”

- How were your questionnaires validated?

- Line 129: replace “used” with “use”

- Cronbach’s alpha of each mentioned scale were at or in most cases well above the 0.7 threshold illustrating strong

consistency and therefore strong scale reliability

- Except for the Self-Stigma of Seeking Help (SSOSH) scale which had a Cronbach’s alpha of 0.667

Statistical Analysis

- Comprehensive statistical analyses were run

- Results of the Hosmer-Lemeshow test can be highly dependent on groups chosen so it is important to acknowledge this in

your manuscript

Results

- Majority of participants (64.5%) were male

- Slight concern that this could lead to results that are not generalizable to the Malaysian population given that males

only comprised 51.4% of its population in 2021 according to Malaysia’s Department of Statistics

- Line 223: add “in” after “diabetes”

- Line 224” add “in” before “20”

Discussion

- Adequately addressed the limitations created by their study sample being majority male

- Line 413-414: rephrase as these lines do not make sense

- Line 431” replace “rises” with “rose”

- The discussion section is a bit disorganized. I suggest revising the order of ideas/references to create better clarity as is

shown in the “Poverty Attribution”, “Stigma and Professional Help-Seeking”, and “Resilience” sections.

Conclusion

- Authors identify a future use for their study results specifically involving the development of psychosocial interventions for

low-income community members.

Tables & Figures

- No comments or suggestions

Reviewer #2: I have the following comments for the authors to address. I am happy to review this paper again.

1) Under the Introduction, the authors stated "This has led to concerns that the COVID-19 pandemic may

94 adversely affect the mental health of populations in LMIC that lack the resources to address

95 the increase in mental health needs of their population". Please refer to the following reference to support this statement:

The impact of COVID-19 pandemic on physical and mental health of Asians: A study of seven middle-income countries in Asia. PLoS One. 2021 Feb 11;16(2):e0246824. doi: 10.1371/journal.pone.0246824. PMID: 33571297.

2) Under the introduction, please discuss the following:

Government response during the pandemic:

Government response moderates the mental health impact of COVID-19: A systematic review and meta-analysis of depression outcomes across countries. J Affect Disord. 2021 May 27;290:364-377. doi: 10.1016/j.jad.2021.04.050. Epub ahead of print. PMID: 34052584.

Worst outcome of COVID infection due to depression

Association Between Mood Disorders and Risk of COVID-19 Infection, Hospitalization, and Death: A Systematic Review and Meta-analysis. JAMA Psychiatry. 2021 Jul 28. doi: 10.1001/jamapsychiatry.2021.1818. Epub ahead of print. PMID: 34319365.

Impact on workers:

Impacts on Occupations During the First Vietnamese National Lockdown. Ann Glob Health. 2020;86(1):112. Published 2020 Sep 3. doi:10.5334/aogh.2976

Impact of lockdown:

Impacts of COVID-19 on the Life and Work of Healthcare Workers During the Nationwide Partial Lockdown in Vietnam. Front Psychol. 2021 Aug 19;12:563193. doi: 10.3389/fpsyg.2021.563193. PMID: 34489769; PMCID: PMC8417359.

3) Under the discussion, the author stated "Younger respondents (less than 30 years old) have a higher risk of developing depression symptoms than older respondents during the pandemic". They should mention the impact on education, examination and graduation as reported in the following study:

Immediate Psychological Responses and Associated Factors during the Initial Stage of the 2019 Coronavirus Disease (COVID-19) Epidemic among the General Population in China. Int J Environ Res Public Health. 2020;17(5):1729. Published 2020 Mar 6. doi:10.3390/ijerph17051729

4) There has been no attempt to interpret the mean Euro-QOL score of this study and compared to other diseases/conditions. In one supplementary file, it seems the mean score is 1 and it should be mentioned in the text and compare to the following conditions so that reader know how it stands. It seems to be higher of the following conditions and it means the QOL is not that bad. Please compare with the following studies under the discussion.

General population under COVID-19 (EuroQol-5D = 0.95) (Tran et al 2020),

patients suffering from diabetes (EuroQol-5D= 0.8) Nguyen Huong Thi Thu et al 2018),

human immunodeficiency virus (HIV) (EuroQol-5D = 0.8) (Tran et al 2018),

skin diseases (EuroQol-5D= 0.73) (Nguyen et al 2019),

respiratory diseases (EuroQol-5D= 0.66) (Ngo et al 2019),

dengue fever (EuroQol-5D= 0.66) (Tran et al 2018),

frail elderly (EuroQol-5D = 0.58) (Nguyen Anh Trung et al 2019)

elderly after fall injury (EuroQol-5D = 0.46) (Vu et al 2019)

fracture injuries (EuroQol-5D = 0.23) (Vu et al 2019).

References:

Tran BX, Nguyen HT, Le HT et al. Impact of COVID-19 on Economic Well-Being and Quality of Life of the Vietnamese During the National Social Distancing. Front Psychol. 2020 Sep 11;11:565153. doi: 10.3389/fpsyg.2020.565153. PMID: 33041928; PMCID: PMC7518066.

Nguyen, Huong Thi Thu et al. “Health-related quality of life in elderly diabetic outpatients in Vietnam.” Patient preference and adherence vol. 12 1347-1354. 27 Jul. 2018, doi:10.2147/PPA.S162892

Tran, Bach Xuan et al. “Depression and Quality of Life among Patients Living with HIV/AIDS in the Era of Universal Treatment Access in Vietnam.” International journal of environmental research and public health vol. 15,12 2888. 17 Dec. 2018, doi:10.3390/ijerph15122888

Nguyen, Sau Huu et al. “Health-Related Quality of Life Impairment among Patients with Different Skin Diseases in Vietnam: A Cross-Sectional Study.” International journal of environmental research and public health vol. 16,3 305. 23 Jan. 2019, doi:10.3390/ijerph16030305

Ngo, Chau Quy et al. “Effects of Different Comorbidities on Health-Related Quality of Life among Respiratory Patients in Vietnam.” Journal of clinical medicine vol. 8,2 214. 7 Feb. 2019, doi:10.3390/jcm8020214

Tran, Bach Xuan et al. “Cost-of-Illness and the Health-Related Quality of Life of Patients in the Dengue Fever Outbreak in Hanoi in 2017.” International journal of environmental research and public health vol. 15,6 1174. 5 Jun. 2018, doi:10.3390/ijerph15061174

Nguyen, Anh Trung et al. “Frailty Prevalence and Association with Health-Related Quality of Life Impairment among Rural Community-Dwelling Older Adults in Vietnam.” International journal of environmental research and public health vol. 16,20 3869. 12 Oct. 2019, doi:10.3390/ijerph16203869

Vu, Hai Minh et al. “Effects of Chronic Comorbidities on the Health-Related Quality of Life among Older Patients after Falls in Vietnamese Hospitals.” International journal of environmental research and public health vol. 16,19 3623. 27 Sep. 2019, doi:10.3390/ijerph16193623

Vu HM, Dang AK, Tran TT, et al Health-Related Quality of Life Profiles among Patients with Different Road Traffic Injuries in an Urban Setting of Vietnam. Int J Environ Res Public Health. 2019 Apr 24;16(8):1462. doi: 10.3390/ijerph16081462. PMID: 31022979; PMCID: PMC6517995.

5) The authors mentioned "These

449 were unexpected findings as females were the most replicable risk factors in the past studies

450 among the general population (85, 86), " Reference 85 and 86 focus on Malaysia. Please refer to the following global study:

Prevalence of Depression in the Community from 30 Countries between 1994 and 2014. Sci Rep. 2018;8(1):2861. Published 2018 Feb 12. doi:10.1038/s41598-018-21243-x

6) The authors stated "Non-communicable diseases, including hypertension and diabetes, were reported by

458 one in three persons within the low-income population". Please refer to the following low income country to support this statement:

Depressive symptoms among elderly diabetic patients in Vietnam. Diabetes Metab Syndr Obes. 2018 Oct 23;11:659-665. doi: 10.2147/DMSO.S179071. PMID: 30425543; PMCID: PMC6204855.

7) The authors stated "Respondents with a known history of hypertension, other diseases and perceived

465 chronic illness as stressful life events likely to report higher depressive symptoms" Please discuss the pathology between depression and heart diseases based on the following:

Factors Associated with the Risk of Developing Coronary Artery Disease in Medicated Patients with Major Depressive Disorder. Int J Environ Res Public Health. 2018 Sep 21;15(10):2073. doi: 10.3390/ijerph15102073. PMID: 30248896; PMCID: PMC6210477.

8) The authors stated "Asthma was associated with a higher risk for anxiety symptoms". This statement needs a reference:

Psychiatric comorbidities in Asian adolescent asthma patients and the contributions of neuroticism and perceived stress. J Adolesc Health. 2014 Aug;55(2):267-75. doi: 10.1016/j.jadohealth.2014.01.007. Epub 2014 Mar 12. PMID: 24630495.

9) Under "Poverty Attribution", please discuss strategies to allow workers to work during the COVID-19 pandemic so that they can have income based on the following study:

Is returning to work during the COVID-19 pandemic stressful? A study on immediate mental health status and psychoneuroimmunity prevention measures of Chinese workforce. Brain Behav Immun. 2020 Jul;87:84-92. doi: 10.1016/j.bbi.2020.04.055. Epub 2020 Apr 23. PMID: 32335200; PMCID: PMC7179503.

10) Under "Stigma and Professional Help-Seeking", please discuss the findings of the following paper that focused on Asia:

Overview of Stigma against Psychiatric Illnesses and Advancements of Anti-Stigma Activities in Six Asian Societies. Int J Environ Res Public Health. 2019 Dec 31;17(1):280. doi: 10.3390/ijerph17010280. PMID: 31906068; PMCID: PMC6981757.

11) Please add a limitation that most factors such as respiratory diseases, marital status, workplace

44 issues, financial constraints, absence of investments, substance use and lack of rental income are not related to COVID-19. The authors should list down factor that should be studied in the future based on the following studies, under future direction.

Exposure to health info/discrimination: The Impact of 2019 Coronavirus Disease (COVID-19) Pandemic on Physical and Mental Health: A Comparison between China and Spain. JMIR Form Res. 2021 Apr 22. doi: 10.2196/27818. Epub ahead of print. PMID: 33900933.

Physical symptoms: The impact of the COVID-19 pandemic on physical and mental health in the two largest economies in the world: a comparison between the United States and China. J Behav Med. 2021 Jun 14:1–19. doi: 10.1007/s10865-021-00237-7. Epub ahead of print. PMID: 34128179; PMCID: PMC8202541.

Face mask use: The Association Between Physical and Mental Health and Face Mask Use During the COVID-19 Pandemic: A Comparison of Two Countries With Different Views and Practices. Front Psychiatry. 2020;11:569981. Published 2020 Sep 9. doi:10.3389/fpsyt.2020.569981

Religion and loss of confidence with doctors: https://www.mdpi.com/2673-5318/2/1/6

Discrimination related to COVID: Psychological impact of COVID-19 pandemic in the Philippines. J Affect Disord. 2020 Aug 24;277:379-391. doi: 10.1016/j.jad.2020.08.043. Epub ahead of print. PMID: 32861839.

Higher number of children in family: Evaluating the Psychological Impacts Related to COVID-19 of Vietnamese People Under the First Nationwide Partial lockdown in Vietnam. Front Psychiatry. 2020 Sep 2;11:824. doi: 10.3389/fpsyt.2020.00824. PMID: 32982807; PMCID: PMC7492529.

12) Under "Stigma and Professional Help-Seeking", please mention internet CBT as it avoid stigma and main mode of psychological treatment during the pandemic:

The most evidence-based treatment is cognitive behaviour therapy (CBT), especially Internet CBT that can prevent the spread of infection during the pandemic.

Use of Cognitive Behavior Therapy (CBT) to treat psychiatric symptoms during COVID-19:

Mental Health Strategies to Combat the Psychological Impact of COVID-19 Beyond Paranoia and Panic. Ann Acad Med Singapore. 2020;49(3):155‐160.

Cost-effectiveness of iCBT:

Moodle: The cost effective solution for internet cognitive behavioral therapy (I-CBT) interventions. Technol Health Care. 2017;25(1):163-165. doi: 10.3233/THC-161261. PMID: 27689560.

Internet CBT can treat psychiatric symptoms such as insomnia:

Efficacy of digital cognitive behavioural therapy for insomnia: a meta-analysis of randomised controlled trials. Sleep Med. 2020 Aug 26;75:315-325. doi: 10.1016/j.sleep.2020.08.020. Epub ahead of print. PMID: 32950013.

6. PLOS authors have the option to publish the peer review history of their article (what does this mean?). If published, this will include your full peer review and any attached files.

Reviewer #1: No

Reviewer #2: No

---

## [Author Response · Author response to Decision Letter 0]

7 Jun 2022

Reviewer #1: 

1. Abstract

- No comments or suggestions

Response: Thank you. 

2. Introduction

- Line 58: replace "suicides" with "suicide": 

Response: Thank you for this. We have amended it accordingly. 

- Line 60: replace "Malaysian" with "Malaysians": 

Response: Thank you. We have made the necessary change (line 60).

- Line 76: remove "on"

Response: Thank you for highlighting this. We have removed the word, "on". 

- Line 91-93: Rephrase as it does not make sense

Response: Apologies for the lack of clarity. The sentence has now been rephrased to A recent study from China showed that resilience scores were inversely associated with severity of mental health symptoms among subjects with mild COVID-19. (Line 102-103)

- Line 99: add "pandemic" after "COVID-19" 

Response: We are grateful to the reviewer for pointing out this oversight. We have now added the word "pandemic" after the "COVID-19" (line 100)

3. Methods

- Manuscript states that the study was conducted from September to November of 2020 in the introduction section but from September to November of 2020 in the methods section. When did this study take place? Clarify and ensure that the information is identical across the manuscript.

Response: Our apologies for the typo errors and confusion. 

Corrections: This was a community-based cross-sectional survey conducted from September 2020 until November 2020(Line 114 -116):

- Alpha value of 0.2 is relatively high, resulting in an increased probability of making a type I error. 

Response: We have made the necessary corrections .

Corrections: The alpha value was 0.05. Epi software with a significance level of 0.05 with a statistical power of 0.8 (Line 118-120)

- Line 130: replace "enrolment" with "enrollment": 

Response: Thank you for highlighting this. We have now corrected it (line 128).

- How were your questionnaires validated?

Response: Thank you for your query. 

“All of the study tools were validated in similar target population but in different location three weeks before the actual data collection. (refer to Table S2). “Cronbach's alpha of each mentioned scale were at or in most cases well above the 0.7 threshold illustrating strong consistency and therefore strong scale reliability.” (Line 201-204)

- Line 129: replace "used" with "use" 

Response: Thank you. We have corrected this (line 146).

- Cronbach's alpha of each mentioned scale was at or in most cases, well above the 0.7 thresholds illustrating strong consistency and, therefore strong scale reliability

Response: Thank you for this reassurance. We have now included this statement in the manuscript (Line 201 to 204) 

- Except for the Self-Stigma of Seeking Help (SSOSH) scale, which had a Cronbach's alpha of 0.667

Response: Thank you for highlighting this. The Cronbach alpha score of 0.667 was obtained by Ibrahim et al in a study conducted on younger age group for similar population. However, validation of the tools in the pilot study revealed excellent Cronbach alpha of 0.84 (Line 187 -188) for adult population. 

Ibrahim, N., Amit, N., Shahar, S., Wee, L.-H., Ismail, R., Khairuddin, R., Siau, C. S., & Safien, A. M. (2019, 2019/06/13). Do depression literacy, mental illness beliefs and stigma influence mental health help-seeking attitude? A cross-sectional study of secondary school and university students from B40 households in Malaysia. BMC Public Health, 19(4), 544. https://doi.org/10.1186/s12889-019-6862-6

4. Statistical Analysis

- Comprehensive statistical analyses were run

Response: Thank you.

- Results of the Hosmer-Lemeshow test can be highly dependent on the groups chosen, so it is important to acknowledge this in your manuscript.

Response: We are grateful to this reviewer for bringing this to our attention. We have now amended the relevant sentences to indicate the following: "To ensure the effectiveness of the Hosmer Lemeshow test, the rule of thumb recommended by Paul et al. (2013) was applied, wherein a study with sample size upto n= 1000, a number of group up to 10 was used" (Line 222-224).

Paul P, Pennell M, Lemeshow S. Standardizing the power of the Hosmer-Lemeshow goodness of fit test in large data sets. Stat Med. 2013;32(1):67-80. DOI: 10.1002/sim.5525

5. Results

Majority of participants (64.5%) were male

- Slight concern that this could lead to results that are not generalizable to the Malaysian population, given that males. Only comprised 51.4% of its population in 2021, according to Malaysia's Department of Statistics

Response: Thank you for highlighting this. Most head of households were male. This is consistent with the distribution of the proportion of males based on data from the Malaysian Household Income Survey (HIS) for low-income households. A couple of similar studies also faced disproportionate in gender's frequencies. In contrast, their studies have more females ( 53% to 70 % more ) (1-3). We did face difficulties engaging some participants due to their long working hours and refusal to participate. This has duly been added in as a limitation of the study: “ The majority of the sample were heads of household, male and older age group, the findings may not be generalisable to all lower-income populations.” (Lines 477-479)

1. Samsudin HB, Nadzrulizam AA. Relationship between B40 Household Income and Demographic Factors in Malaysia. Int J Eng Innov Technol. 2020;10(2).

2. Lugova H, Andoy-Galvan JA, Patil SS, Wong YH, Baloch GM, Suleiman A, et al. Prevalence and Associated Factors of the Severity of Depression, Anxiety and Stress Among Low-Income Community-Dwelling Adults in Kuala Lumpur, Malaysia. Community mental health journal. 2021.

3. Abd Rashid R, Kanagasundram S, Danaee M, Abdul Majid H, Sulaiman A, Ahmad Zahari M, et al. The Prevalence of Smoking, Determinants and Chance of Psychological Problems among Smokers in an Urban Community Housing Project in Malaysia. International journal of environmental research and public health. 2019;16(10).

- Line 223: add "in" after "diabetes"

Response: Thank you. We have added "in". (Line 242)

- Line 224" add "in" before "20"

Response: Thank you. We have amended the sentence accordingly (Line 243).

6. Discussion

- Adequately addressed the limitations created by their study sample being majority male

- Line 413-414: rephrase as these lines do not make sense

Response: Apologies for the lack of clarity. We have now amended the relevant sentence to, "Survey on Alcohol and Related Conditions, has shown (NESARC) (91) anxiety does not significantly association with quality of life" (Line 334-336).

- Line 431" replace "rises" with "rose" 

Response: Thank you for your suggestion. We have now made the relevant replacement (Line 355)

- The discussion section is a bit disorganized. I suggest revising the order of ideas/references to create better clarity as is shown in the "Poverty Attribution", "Stigma and Professional Help-Seeking", and "Resilience" sections.

Response: We do apologize for being disorganized and have taken on board the suggestions by this reviewer. The section has now been reorganized, and changes are highlighted in the tracked changes version (Line 420-470)

7. Conclusion

- Authors identify a future use for their study results, specifically involving the development of psychosocial interventions for low-income community members.

Response: Thank you for these encouraging words. We can't agree more. 

8. Tables & Figures

- No comments or suggestions

Response: Thank you.

Reviewer #2: I have the following comments for the authors to address. I am happy to review this paper again.

1. Under the Introduction, the authors stated "This has led to concerns that the COVID-19 pandemic may adversely affect the mental health of populations in LMIC that lack the resources to address the increase in mental health needs of their population". Please refer to the following reference to support this statement:

Wang C, Tee M, Roy A, Fardin M, Srichokchatchawan W, Habib H, et al. The impact of COVID-19 pandemic on physical and mental health of Asians: A study of seven middle-income countries in Asia. PLoS ONE. 2021;16(2):e0246824-e.

Response: We are most grateful to this reviewer for his suggestion. We have now added in the recommended reference. "This has led to concerns that the COVID-19 pandemic may adversely affect the mental health of populations in LMIC (34) that lack the resources to address the increase in mental health needs of their population (34, 35)". (Lines 104-106)

2. Under the introduction, please discuss the following:

Government response during the pandemic:

Government response moderates the mental health impact of COVID-19: A systematic review and meta-analysis of depression outcomes across countries. J Affect Disord. 2021 May 27;290:364-377. doi: 10.1016/j.jad.2021.04.050. Epub ahead of print. PMID: 34052584.

Worst outcome of COVID infection due to depression

Association Between Mood Disorders and Risk of COVID-19 Infection, Hospitalization, and Death: A Systematic Review and Meta-analysis. JAMA Psychiatry. 2021 Jul 28. doi: 10.1001/jamapsychiatry.2021.1818. Epub ahead of print. PMID: 34319365.

Impact on workers:

Impacts on Occupations During the First Vietnamese National Lockdown. Ann Glob Health. 2020;86(1):112. Published 2020 Sep 3. doi:10.5334/aogh.2976

Impact of lockdown:

Impacts of COVID-19 on the Life and Work of Healthcare Workers During the Nationwide Partial Lockdown in Vietnam. Front Psychol. 2021 Aug 19;12:563193. doi: 0.3389/fpsyg.2021.563193. PMID: 34489769; PMCID: PMC8417359.

Response: Thank you for all the useful suggestions. We have incorporated this reviewer's suggestions in the relevant sections (lines 88-99). 

" The recent COVID-19 pandemic has crippled the economy and heightened the pre-existing financial strain among low-income populations. Undoubtedly, the government's containment measures prevented the spread of the coronavirus, but a significant number of people had suffered from financial loss due to job loss and pay cuts (28). Unemployment and financial issues are important stressors that predict depression (29). Even though debatable findings from a multi-national meta-analysis showed that stringent government containment measures represents a protective factor for depression (30), this should not delay any public health measures that prioritise early detection of at-risk individuals for mental health issues with timely effective interventions to mitigate any potential negative consequences of the COVID-19 pandemic on mental health as well as effects of mental health on COVID-19 severity. A recent meta-analysis suggests that individuals with pre-existing mood disorders are at higher risk of COVID-19 hospitalisation and death (31).” (lines 88-99).

3. Under the discussion, the author stated "Younger respondents (less than 30 years old) have a higher risk of developing depression symptoms than older respondents during the pandemic". They should mention the impact on education, examination and graduation as reported in the following study:

Immediate Psychological Responses and Associated Factors during the Initial Stage of the 2019 Coronavirus Disease (COVID-19) Epidemic among the General Population in China. Int J Environ Res Public Health. 2020;17(5):1729. Published 2020 Mar 6. doi:10.3390/ijerph17051729

Chen T, Lucock M. The mental health of university students during the COVID-19 pandemic: An online survey in the UK. PLoS ONE. 2022;17(1):e0262562.

Wang C, Pan R, Wan X, Tan Y, Xu L, Ho CS, et al. Immediate Psychological Responses and Associated Factors during the Initial Stage of the 2019 Coronavirus Disease (COVID-19) Epidemic among the General Population in China. International journal of environmental research and public health. 2020;17(5):1729.

Response: We are grateful to this reviewer for the, once again, excellent suggestions. The lockdown and social distancing measures instituted by the government to reduce transmission had indeed led to prolonged school closures and pivoting to online teaching. Indeed those within the local socioeconomic classes were more badly affected due to the limited access to good internet connections as well as electronic devices. Consequently we have made the following amendments: 

Several community-based studies revealed high prevalence of mental health problems among young adults especially among students (104, 105). Prolonged school closures and a switch to online learning occurred as a result of lockdown and social distancing measures. Those within the local socioeconomic classes were disadvantaged as a result of limited access to good internet connections and electronic devices (105). (Lines 368 to 372)

4. There has been no attempt to interpret the mean Euro-QOL score of this study and compared to other diseases/conditions. In one supplementary file, it seems the mean score is 1 and it should be mentioned in the text and compared to the following conditions so that the reader knows how it stands. It seems to be higher of the following conditions and it means the QOL is not that bad. Please compare with the following studies under the discussion.

General population under COVID-19 (EuroQol-5D = 0.95) (Tran et al 2020)

patients suffering from diabetes (EuroQol-5D= 0.8) Nguyen Huong Thi Thu et al 2018),

human immunodeficiency virus (HIV) (EuroQol-5D = 0.8) (Tran et al 2018),

skin diseases (EuroQol-5D= 0.73) (Nguyen et al 2019),

respiratory diseases (EuroQol-5D= 0.66) (Ngo et al 2019),

dengue fever (EuroQol-5D= 0.66) (Tran et al 2018),

frail elderly (EuroQol-5D = 0.58) (Nguyen Anh Trung et al 2019)

elderly after fall injury (EuroQol-5D = 0.46) (Vu et al 2019)

fracture injuries (EuroQol-5D = 0.23) (Vu et al 2019).

References:

Tran BX, Nguyen HT, Le HT et al. Impact of COVID-19 on Economic Well-Being and Quality of Life of the Vietnamese During the National Social Distancing. Front Psychol. 2020 Sep 11;11:565153. doi: 10.3389/fpsyg.2020.565153. PMID: 33041928; PMCID: PMC7518066.

Nguyen, Huong Thi Thu et al. "Health-related quality of life in elderly diabetic outpatients in Vietnam." Patient preference and adherence vol. 12 1347-1354. 27 Jul. 2018, doi:10.2147/PPA.S162892

Tran, Bach Xuan et al. "Depression and Quality of Life among Patients Living with HIV/AIDS in the Era of Universal Treatment Access in Vietnam." International journal of environmental research and public health vol. 15,12 2888. 17 Dec. 2018, doi:10.3390/ijerph15122888

Nguyen, Sau Huu et al. "Health-Related Quality of Life Impairment among Patients with Different Skin Diseases in Vietnam: A Cross-Sectional Study." International journal of environmental research and public health vol. 16,3 305. 23 Jan. 2019, doi:10.3390/ijerph16030305

Ngo, Chau Quy et al. "Effects of Different Comorbidities on Health-Related Quality of Life among Respiratory Patients in Vietnam." Journal of clinical medicine vol. 8,2 214. 7 Feb. 2019, doi:10.3390/jcm8020214

Tran, Bach Xuan et al. "Cost-of-Illness and the Health-Related Quality of Life of Patients in the Dengue Fever Outbreak in Hanoi in 2017." International journal of environmental research and public health vol. 15,6 1174. 5 Jun. 2018, doi:10.3390/ijerph15061174

Nguyen, Anh Trung et al. "Frailty Prevalence and Association with Health-Related Quality of Life Impairment among Rural Community-Dwelling Older Adults in Vietnam." International journal of environmental research and public health vol. 16,20 3869. 12 Oct. 2019, doi:10.3390/ijerph16203869

Vu, Hai Minh et al. "Effects of Chronic Comorbidities on the Health-Related Quality of Life among Older Patients after Falls in Vietnamese Hospitals." International journal of environmental research and public health vol. 16,19 3623. 27 Sep. 2019, doi:10.3390/ijerph16193623

Vu HM, Dang AK, Tran TT, et al Health-Related Quality of Life Profiles among Patients with Different Road Traffic Injuries in an Urban Setting of Vietnam. Int J Environ Res Public Health. 2019 Apr 24;16(8):1462. doi: 10.3390/ijerph16081462. PMID: 31022979; PMCID: PMC6517995.

Response: Thank you for the extensive recommendations which we have taken into account wholeheartedly. In response to the above highlighted points, we have made the following extensive amendments:

Response: The data for EQ-5D and Utility scores are highly skewed even after transformation. Which led to the presentation of median scores. However, to enable comparisons with other studies, we have included mean scores (line 273-274) 

However, the assumption of normal distribution required for linear regression was violated, the data was dichotomised. We apologize if this was not made clear within our manuscript. We have now added the following statement within the manuscript: "The variable EQ-5D was dichotomised into no problem with QoL =0 and problem with QoL = 1, using the cut-off score of 1.00" (Line 209-211 )

Here are the changes we made:

“The EQ-5D has an overall median score of 1.00 (0.8) and a mean score of 0.94 (SD: 0.12)”. (line 272-273)

The EQ-5D mean score of 0.94 from the current study is comparable to studies done by Tran et al (2020) and Vu et al (2020) among the general population in Vietnam during COVID-19 pandemic (77,78). A higher proportion of participants had full health scoring (72.2 %) compared to the participants from Vietnam (54.9 % to 60.0%)(78). The mean score of EQ-5D was also found to be better than those studies which focused on chronic diseases in Malaysia and other countries , which entails patients suffering from diabetes (2, 79), human immunodeficiency virus (HIV) (80), skin diseases (81), respiratory diseases (82) dengue fever (83), frail elderly (84), elderly after fall injury (85) fracture injuries (86) and Chronic Myeloid Leukemia (87)”. (lines 313 to 320)

5. The authors mentioned "These were unexpected findings as females were the most replicable risk factors in the past studies among the general population (85, 86), " Reference 85 and 86 focus on Malaysia. Please refer to the following global study:

Prevalence of Depression in the Community from 30 Countries between 1994 and 2014. Sci Rep. 2018;8(1):2861. Published 2018 Feb 12. doi:10.1038/s41598-018-21243-x

Response: We are grateful to this reviewer for your suggestions. We have added in the relevant reference accordingly, "Gender did not reveal any positive findings for mental health status in this study. These were unexpected findings as females were the most replicable risk factors in the past studies among the general population(3, 106, 107)…". (Line 373-375)

6. The authors stated "Non-communicable diseases, including hypertension and diabetes, were reported by one in three persons within the low-income population". Please refer to the following low income country to support this statement:

Depressive symptoms among elderly diabetic patients in Vietnam. Diabetes Metab Syndr Obes. 2018 Oct 23;11:659-665. doi: 10.2147/DMSO.S179071. PMID: 30425543; PMCID: PMC6204855.

Response: Thank you once again for the recommendation. We have added in the relevant reference. "The prevalence of the depression in this study is comparatively higher than the NHMS 2019 but lower than similar studies in the LMICs (9,110)" (Lines 383 to 385).

7. The authors stated "Respondents with a known history of hypertension, other diseases and perceived chronic illness as stressful life events likely to report higher depressive symptoms" Please discuss the pathology between depression and heart diseases based on the following:

Factors Associated with the Risk of Developing Coronary Artery Disease in Medicated Patients with Major Depressive Disorder. Int J Environ Res Public Health. 2018 Sep 21;15(10):2073. doi: 10.3390/ijerph15102073. PMID: 30248896; PMCID: PMC6210477.

Response: We are grateful to the reviewer for bringing this to our attention. We have now made the following adjustments, " The mechanisms underlying the causal relationship between mental and physical health are multifactorial that entail biological, psychosocial, environmental and behavioural. Environmental factors that induced chronic stress (psychosocial risk factors), may promote physical illness such as obesity due to unhealthy eating and a sedentary lifestyle. Through the psychosocial pathway, the physiological feedback from environmental stress factors results in the production of intermediate markers like pro-inflammatory markers (interleukin (IL)–6 and tumor necrosis factor alpha (TNF–α) (116), vascular stiffening and endothelial dysfunction leading to cardiovascular outcomes (117). In obesity , high visceral fat is the major site for deposition of (IL)-6 which explain the association between depression, inflammation, metabolic risk factors and cardiovascular diseases (113,118). Therefore, the long-standing financial strain concurrent with the occurrence of non-communicable disease may contribute to the mental health problems experienced by the B40 community, and was explained by the psychosocial risk factors model”. (lines 394 to 406).

8. The authors stated "Asthma was associated with a higher risk for anxiety symptoms". This statement needs a reference:

Psychiatric comorbidities in Asian adolescent asthma patients and the contributions of neuroticism and perceived stress. J Adolesc Health. 2014 Aug;55(2):267-75. doi: 10.1016/j.jadohealth.2014.01.007. Epub 2014 Mar 12. PMID: 24630495.

Response: Thank you for the recommended reference which has now been added in. “Asthma was associated with a higher risk for anxiety symptoms (111,112)”. (Lines 390)

9. Under "Poverty Attribution", please discuss strategies to allow workers to work during the COVID-19 pandemic so that they can have income based on the following study:

Is returning to work during the COVID-19 pandemic stressful? A study on immediate mental health status and psychoneuroimmunity prevention measures of Chinese workforce. Brain Behav Immun. 2020 Jul;87:84-92. doi: 10.1016/j.bbi.2020.04.055. Epub 2020 Apr 23. PMID: 32335200; PMCID: PMC7179503.

Response: We are grateful once more for the excellent recommendation. The following amendment has now been made: " While allowing workers to return to work during a pandemic eased the financial strain, this conversely increased infection risk. Studies suggest that implementation of psychoneuroimmunological preventive measures at the workplace which entails practising hand hygiene, maintaining social distancing, wearing facemask and healthy lifestyles, may ease the psychological distress and smoothen the process of return to work (133, 134).". (Lines 457 to 462)

10. Under "Stigma and Professional Help-Seeking", please discuss the findings of the following paper that focused on Asia:

Overview of Stigma against Psychiatric Illnesses and Advancements of Anti-Stigma Activities in Six Asian Societies. Int J Environ Res Public Health. 2019 Dec 31;17(1):280. doi: 10.3390/ijerph17010280. PMID: 31906068; PMCID: PMC6981757.

Response: Thank you for the suggested reference alongside the useful comment. We have now made the following changes in the relevant section, " The Let’s Talk campaign was launched recently as part of health a promotion from Malaysia’s Ministry of Health, which aimed to destigmatise mental illness through public education and encourage people to seek help. However, more well-defined with areas of focus, evidence-based strategies are still needed. For instance, Japan and Hong Kong have standardised and advocated the used of less stigmatised terms, while China has enacted a national legislation and public education emphasizing the need to respect psychiatric patients as an anti-discriminatory approach (127)”. (lines 434-440)"

11. Please add a limitation that most factors such as respiratory diseases, marital status, workplace 44 issues, financial constraints, absence of investments, substance use and lack of rental income are not related to COVID-19. The authors should list down factor that should be studied in the future based on the following studies, under future direction.

Exposure to health info/discrimination: The Impact of 2019 Coronavirus Disease (COVID-19) Pandemic on Physical and Mental Health: A Comparison between China and Spain. JMIR Form Res. 2021 Apr 22. doi: 10.2196/27818. Epub ahead of print. PMID: 33900933.

Physical symptoms: The impact of the COVID-19 pandemic on physical and mental health in the two largest economies in the world: a comparison between the United States and China. J Behav Med. 2021 Jun 14:1–19. doi: 10.1007/s10865-021-00237-7. Epub ahead of print. PMID: 34128179; PMCID: PMC8202541.

Face mask use: The Association Between Physical and Mental Health and Face Mask Use During the COVID-19 Pandemic: A Comparison of Two Countries With Different Views and Practices. Front Psychiatry. 2020;11:569981. Published 2020 Sep 9. doi:10.3389/fpsyt.2020.569981

Religion and loss of confidence with doctors: https://www.mdpi.com/2673-5318/2/1/6

Discrimination related to COVID: Psychological impact of COVID-19 pandemic in the Philippines. J Affect Disord. 2020 Aug 24;277:379-391. doi: 10.1016/j.jad.2020.08.043. Epub ahead of print. PMID: 32861839.

Higher number of children in family: Evaluating the Psychological Impacts Related to COVID-19 of Vietnamese People Under the First Nationwide Partial lockdown in Vietnam. Front Psychiatry. 2020 Sep 2;11:824. doi: 10.3389/fpsyt.2020.00824. PMID: 32982807; PMCID: PMC7492529.

Response: Thank you for the above extensive recommendations, which we have addressed in its entirety. The following changes has been made as a results:

"In addition, most of the studied factors were not related to COVID-19, hence the future research direction should focus on the effect of physical symptoms (137), facemask use (138), discrimination related to COVID-19 positive cases (139), higher numbers of children in the family (140), cross-cultural belief or religiosity (141), and impact of excessive exposure of the COVID-19 related health information on mental health (142). (lines 480-483) "

12. Under "Stigma and Professional Help-Seeking", please mention internet CBT as it avoid stigma and main mode of psychological treatment during the pandemic:

The most evidence-based treatment is cognitive behaviour therapy (CBT), especially Internet CBT that can prevent the spread of infection during the pandemic.

Use of Cognitive Behavior Therapy (CBT) to treat psychiatric symptoms during COVID-19:

Mental Health Strategies to Combat the Psychological Impact of COVID-19 Beyond Paranoia and Panic. Ann Acad Med Singapore. 2020;49(3):155‐160.

Cost-effectiveness of iCBT:

Moodle: The cost-effective solution for internet cognitive behavioral therapy (I-CBT) interventions. Technol Health Care. 2017;25(1):163-165. doi: 10.3233/THC-161261. PMID: 27689560.

Internet CBT can treat psychiatric symptoms such as insomnia:

Efficacy of digital cognitive behavioural therapy for insomnia: a meta-analysis of randomised controlled trials. Sleep Med. 2020 Aug 26;75:315-325. doi: 10.1016/j.sleep.2020.08.020. Epub ahead of print. PMID: 32950013.

Response: Thank you for the above extensive recommendations. The following changes has been " To date, internet-based cognitive behavioral therapy (CBT) is the most documented evidence-based, effective in psychiatric symptoms alleviation (128) and cost-effective (129) psychological intervention, which has been embraced into the mainstream to address the mental health burden during the pandemic (130)." (line 440-443).

---

## [Decision Letter · Decision Letter 1]

7 Jul 2022

PONE-D-22-05096R1Psychosocial factors associated with mental health and quality of life during the COVID-19 pandemic among low-income urban dwellers in Peninsular Malaysia

Psychosocial factors for mental health among the low-income communityPLOS ONE

Dear,

Thank you for submitting your manuscript to PLOS ONE. After careful consideration, we feel that it has merit but does not fully meet PLOS ONE’s publication criteria as it currently stands. Therefore, we invite you to submit a revised version of the manuscript that addresses the points raised during the review process. Please submit your revised manuscript by Aug 21 2022 11:59PM If you will need more time than this to complete your revisions, please reply to this message or contact the journal office at plosone@plos.org. Please include the following items when submitting your revised manuscript:A rebuttal letter that responds to each point raised by the academic editor and reviewer(s). You should upload this letter as a separate file labeled 'Response to Reviewers'.A marked-up copy of your manuscript that highlights changes made to the original version. You should upload this as a separate file labeled 'Revised Manuscript with Track Changes'.An unmarked version of your revised paper without tracked changes. You should upload this as a separate file labeled 'Manuscript'.If applicable, we recommend that you deposit your laboratory protocols in protocols.io to enhance the reproducibility of your results. Protocols.io assigns your protocol its own identifier (DOI) so that it can be cited independently in the future. For instructions see: https://journals.plos.org/plosone/s/submission-guidelines#loc-laboratory-protocols. Additionally, PLOS ONE offers an option for publishing peer-reviewed Lab Protocol articles, which describe protocols hosted on protocols.io. Read more information on sharing protocols at https://plos.org/protocols?utm_medium=editorial-email&utm_source=authorletters&utm_campaign=protocols.

We look forward to receiving your revised manuscript.

Kind regards,

Muhammad Shahzad Aslam, Ph.D.,M.Phil., Pharm-D

Academic Editor

PLOS ONE

Journal Requirements:

Reviewers' comments:

Reviewer's Responses to Questions

**Comments to the Author**

1. If the authors have adequately addressed your comments raised in a previous round of review and you feel that this manuscript is now acceptable for publication, you may indicate that here to bypass the “Comments to the Author” section, enter your conflict of interest statement in the “Confidential to Editor” section, and submit your "Accept" recommendation.

Reviewer #2: All comments have been addressed

Reviewer #3: (No Response)

Reviewer #4: All comments have been addressed

2. Is the manuscript technically sound, and do the data support the conclusions?

Reviewer #2: Yes

Reviewer #3: Yes

Reviewer #4: Yes

3. Has the statistical analysis been performed appropriately and rigorously? 

Reviewer #2: Yes

Reviewer #3: Yes

Reviewer #4: Yes

4. Have the authors made all data underlying the findings in their manuscript fully available?

Reviewer #2: Yes

Reviewer #3: Yes

Reviewer #4: Yes

5. Is the manuscript presented in an intelligible fashion and written in standard English?

Reviewer #2: Yes

Reviewer #3: Yes

Reviewer #4: Yes

6. Review Comments to the Author

Reviewer #2: I recommend the study "Psychosocial factors associated with mental health and quality of life during the

COVID-19 pandemic among low-income urban dwellers in Peninsular Malaysia

Psychosocial factors for mental health among the low-income community" for publication.

Reviewer #3: Data availability statement: Data cannot be shared publicly because of the vulnerability of the population. Was data anonymised? How was the data stored and managed? De-identified data would not compromise the subjects.

Minor comments attached.

Reviewer #4: I was wondering during the MCO (September-November 2020) when the research was done face to face, were the researches welcomed to the subjects' homes or was there any resistance faced? As we know, at that point, movements were still restricted so was it ethical to conduct this study face-to face? Would it not have increased the risk of infection? How was this addressed?

7. PLOS authors have the option to publish the peer review history of their article (what does this mean?). If published, this will include your full peer review and any attached files.

Reviewer #2: No

Reviewer #3: No

Reviewer #4: **Yes: **Aida Syarinaz Ahmad Adlan

---

## [Author Response · Author response to Decision Letter 1]

17 Jul 2022

16th July 2022

Associate Professor Dr Mas Ayu Said

Unit Epidemiology,

Department of Social and Preventive Medicine,

Faculty of Medicine,

University of Malaya,

50603 Kuala Lumpur, Malaysia

Emily Chenette, Editor-In-Chief

PLOS

1265 Battery Street, Suite 200

San Francisco, CA 94111

United States

Dear Editors, 

Manuscript Resubmission with Rebuttal Responses: Psychosocial factors associated with mental health and quality of life during covid-19 pandemic among low-income urban dwellers in Peninsular Malaysia

Greetings from University Malaya, Malaysia Research University Networking (MRUN). We would like to resubmit our manuscript entitled "Psychosocial factors associated with mental health and quality of life during covid-19 pandemic among low-income urban dwellers in Peninsular Malaysia", written by Wong Min Fui @ Esther Wong, Mas Ayu Said, Hazreen, Abdul Majid, Rozmi Ismail and Tan Maw Pin. All authors have approved the manuscript and agree with its submission to Plos One. 

This manuscript's contents have not been copyrighted, submitted, or published elsewhere while awaiting publication in this journal. There is also no conflict of interest to disclose. 

We thank the editor and the reviewers for their insightful comments on our manuscript. Below are our responses to each point the academic editor and reviewers raised. We hope that we satisfyingly addressed them and that the manuscript will be now suited for publication.

Thank you for your consideration and time in reviewing our submission. We look forward to hearing from you.

Yours sincerely, 

Associate Professor Dr Mas Ayu Said 

Corresponding Authors 

Principal Investigator of MRUN

Department of Social and Preventive Medicine, 

University of Malaya, Malaysia

mas@ummc.um.edu.my

Academic Editor

Journal Requirements:

Please review your reference list to ensure that it is complete and correct. If you have cited papers that have been retracted, please include the rationale for doing so in the manuscript text, or remove these references and replace them with relevant current references. Any changes to the reference list should be mentioned in the rebuttal letter that accompanies your revised manuscript. If you need to cite a retracted article, indicate the 'article's retracted status in the References list and include a citation and full reference for the retraction notice.

Response: 

The reference list was cross-checked and updated. New references were added as below : 

48. Mohamad AS, Draman S, Aris MAM, Musa R, Rus RM, Malik M. Depression, anxiety, and stress among adolescents in Kuantan and its association with religiosity-a pilot study. IIUM Medical Journal Malaysia. 2018;17(2):92-5.

Reviewers 

Reviewer #2: I recommend the study "Psychosocial factors associated with mental health and quality of life during the

COVID-19 pandemic among low-income urban dwellers in Peninsular Malaysia

Psychosocial factors for mental health among the low-income community" for publication.

- Response : Thank you for your suggestion.

Reviewer #3: Data availability statement: Data cannot be shared publicly because of the vulnerability of the population. Was data anonymised? How was the data stored and managed? De-identified data would not compromise the subjects.

Minor comments attached.

- Response: Thank you for your questions. 

The data is anonymised and stored according to the standard protocol stipulated in the data management and storage guidelines from University Malaya. The data will be available upon request by other interested researchers. 

Reviewer #4: I was wondering during the MCO (September-November 2020) when the research was done face to face, were the research welcomed to the subjects' homes or was there any resistance faced? As we know, at that point, movements were still restricted so was it ethical to conduct this study face-to face? Would it not have increased the risk of infection? How was this addressed?

- Response: Thank you for your questions. September to early November 2020 were within the period of Recovery of Movement Control Order (RMCO) in Malaysia. As COVID-19 cases were successful under control in this country, the SOP was relaxed, giving the window of opportunity for the research team to conduct the face-to-face data collection with permission from the Faculty of Medicine. Therefore, the team did not face many issues during the data collection. Nevertheless, the Principle Investigator had obtained approval from the Dean's office of the Faculty of Medicine to conduct the data collection strictly with COVID-19 SOP. Enumerators were given a proper briefing on the fieldwork COVID-19 protocol from the University Malaya before the commencement of the data collection. 

 

Responses to the email attachments (Words Documents)

Reviewer # xx

1. Abstract. 

a) Conclusion. I don't think using words 'higher proportion' compared to 'pre-pandemic reports' can be used in conclusions. There is no information on pre-pandemic reports in the background or results. Therefore, if a reader is only looking through the abstract without reading the whole paper, they would not come to this conclusion. 

- Response: Thank you for your response. The statement has been revised as below:

“A high proportion of mild to severe anxiety and depression symptoms was reported in the sampled urban poor population.”(Line 51 -52)

Track-changes document 

2. Introduction

a) Lines 63-64. Authors state the increase in suicide cases from 609 in 2019 to 631 in 2020. Was there also an increase in 'country's population (and thus no increase in suicide rates per capita) over that time? 

- Response: Thank you for your comments. We have implemented the changes as below:

“However, since there was an increase in country's population, thus no increase in suicide rates per capita was observed.” (Line 63-65)

b) Line 85. Please provide description or an example of 'structural 'issues'.

- Response: Thank you for your responses, based on the reference by Mickelson et al. 2014, the description of 'structural 'issues' has been added into the statement: 

"Additionally, low-income individuals' perceptions of the structural issues, attributions related to government blame or discrimination, may subsequently influence their decision-making and mental health.”(Line 84-86)

c) Line 92. 'Financial 'loss' and 'job 'loss' used in the same line. Suggest changing wording to avoid repetition.

- Response: Thank you for your comments. We have implemented the changes as below:

"Undoubtedly, the government's containment measures prevented the spread of the coronavirus, but a significant number of people suffered from financial loss due to job dismissal and pay cuts.” (Line 90-92)

d) Line 93. ''…financial issues are important stressors that predict 'depression' – suggest change wording to 'can lead 'to' or similar.

- Response: Thank you for your comments. We have implemented the changes as below:

"Unemployment and financial issues are important stressors that can lead to depression.” (Line 92-93)

e) Lines 93 – 96. Very long and confusing sentence. Suggest you split it in two.

- Response: Improvement has been done as shown below:

"Even though evidence showed stringent government containment measures moderate depression by promoting trust and ease uncertainty (30), timely implementation of early screening and intervention for mental health at-risk individuals should be in place to prevent severe COVID-19 cases. A recent meta-analysis suggests that individuals with pre-existing mood disorders are at higher risk of COVID-19 hospitalisation and death (31)."(Line 93-96).

f) Line 95. ''…stringent government containment measures represents a protective factor for 'depression' – 'what's the mechanism? How is it a protective factor? (reduces fear of getting sick and being hospitalised?)

- Response: Improvement has been done and shown below:

"Even though evidence showed stringent government containment measures moderate depression by promoting trust and ease uncertainty (30), timely implementation of early screening and intervention for mental health at-risk individuals should be in place to prevent severe COVID-19 cases.”(Line 93-94

g) Line 95. ''…this should not delay any public…'' What is '''this' referring to?

- Response: Kindly refer to the highlighted statement shown below:

"Even though evidence showed stringent government containment measures moderate depression by promoting trust and ease uncertainty (30), timely implementation of early screening and intervention for mental health at-risk individuals should be in place to prevent severe COVID-19 cases.”(Line 94-95)

 

3. Methods

a) Line 138. How the data on psychiatric illness was obtained?

- Response: Thank you for the question.

“ The history of psychiatric illness was a self-reporting response under the sub-heading of the presence of chronic illness (health domain) in the socio-demographic survey form.”(line 142-144) 

b) Line 139. The whole household where non-Malaysians lived was excluded?

- Response: Thank you for the question. 

The enumerators have made sure no Malaysian was in the respective house before they exclude the whole household. 

c) Line 144. ''…Since this is a face-to-face…'' – change to was. Keep your past and present tense consistent throughout. 

- Response: Thank you for the reminder. The correction has been implemented:

“Since this was a face-to-face data collection, each enumerator was briefed about the University of Malaya COVID-19 Fieldwork Safety Protocol.”(line 129-130)

4. Results

a) Line 247-248. Of 504 eligible participants, 432 (82.7%) completed the survey. 

- Response: Correction has been done. “Of 504 total eligible participants, 432 (82.7%) completed the survey.”(line 231)

b) Table 1. 

I cannot find household income before Covid-19 in this table.

- Response: Apologies for the lack of information on the pre-Covid-19 income. Now we have added the information in the Table 1.

Secondary schools and more than secondary schools % are in brackets. Is there a reason why this is different to other data presented in this table?

- Response: Thank you for the question. Our apologies, we have overlooked, and we have removed the brackets.

Employed: yes/no or employment status: employed/not employed. 'Employment 'status' does not define if you are or are not employed. 

- Response: The questionnaire is in Malay language, and participants were required to indicate whether if they are currently employed or not. Therefore, we have amended the table input as below:

“Are you currently employed or working (Before COVID-19)? “

“Are you currently employed or working (After COVID-19)?”

Employment status (during Covid-19). Change the order of yes and no lines, so it matches the order of the employment status (before Covid-19).

- Response: Our apologies, and we have amended the input. 

Household size. =4 belongs to which bracket?

Household size (person)

- Response: Our apologies and the shortcoming have been rectified:

Less than 4 (<4)

More than or same as 4 (≥4)

House ownership. What is inherited mean? 'Aren't they home owners?

- Response: Thank you for your questions. Below is our explanation: 

"Home owners" mean they are the first owner and may have still served the house loan, whereas inherited means it is property passed down from their ancestor, and they are part of the ownership sharing with their siblings. This reflects their socio-economic status". 

c) Lines 256-257. Did you mean 'Structural and socio-economic domains scores of poverty attribution were…'? It reads as if you are providing scores for three domains.

- Response: Thank you for the question. We have rephrased the statement: 

"Poverty attribution of structural (3.6 (SD 0.8)) and socio-economic (3.8 (SD 0.9)) domains achieved the highest total mean scores out of the four domains of PA-21 (line 238-240)".

d) Lines 262 – 264. Not clear. Is this 30% of male respondents or 30% of all participants? 130 is a third of your entire cohort, but this reads as if you are talking about male respondents only. n=278 male 130/278 (xxx%) used substances

- Response: Apology for the typo. The statement was rephrased:

“Among 130 (30.2%) respondents, substance use was present, with tobacco, alcohol, and sleeping pills being the most used substances. .”(line 247-248) 

e) Line 264-265. Is this sentence still describing male participants only?

- Response: Apology for the typo. The statement was rephrased:

“Substance use was present in 130 (30.2%) respondents, with tobacco, alcohol, and sleeping pills being the most used substances .”(line 245-246)

f) Line 266. ''(n=30)'' Is this a typo?

- Response: Apology for the typo. The shortfall was rectified.

g) Line 272-279. age less than 30... is a variable 'those aged less than '30' refers to participants. You talking about participants not variables here. This applies to first two sentences in this paragraph. Try to re-phrase if you want to write about factors.

- Response: Thank you for the responses. We have rephrased the statement:

“Factors positively associated with mild to severe symptoms of depression entail the age group of "less than 30 years" OR 5.11 (95%CI 2.04, 12.83), self-reported hypertension, having other chronic illnesses, and having the presence of past stressful life events (physical assault, long term illness, family issues and workplace issues)” (Line 254-257).

h) Table 2. Sleep pill usage. 'Reference' duplication. Describe all the abbreviations used in this table. 

- Response: Thank you for the reminder. Corrections have been done in Table 2.

 

5. Discussion

a) Line 319-320. Suggest use 'respondents' here instead of group and delete word 'respondents' after 'financial strain'.

- Response: Thank you for the suggestion. We have edited the sentence. 

"Lower-income respondents with higher financial strain were more likely to experience mild to severe depression (28).”(Line 298-299)

b) Line 332. It would be good to see some numbers found in studies you're including in your reference index to support your statement.

- Response: Thank you for the suggestion. We have edited the sentence.

“The overall prevalence for poorer quality of life was 27.8%, comparable to a validation study from Trinidad and Tobago (28.0%) (71) but lower than the majority of the pre-pandemic studies from other countries (EQ-5D: 54.0%-69.7%) (72-75). The figure remain low even after accounting for socio-demographic and chronic illness characteristics (72,76).” (line 311-315)

c) Line 338. Delete 'were'.

- Response: The changes have been done.

d) Line 388.'...to the mid-20s to the 30s of age' – fix wording

- Response: Correction has been made: "…20 to 30-years of age…" at line 367

e) Line 434. Cardiovascular problems? Adverse cardiovascular outcomes? Use something similar instead just 'cardiovascular outcomes'.

- Response: Thank you for the suggestion. We have made the changes: 

"…vascular stiffening and endothelial dysfunction leading to adverse cardiovascular outcomes (117)." (Line 406)

f) Line 466. Not sure what you mean by 'anxiety symptoms but not for depression'

- Response: Thank you for the response. Here is the improved version: 

“The final analysis revealed, those with higher self-stigma scores were likely to have anxiety symptoms, whereas lower mental health-seeking attitudes predicted higher depressive symptoms. Poor help-seeking attitude prevents the low-income respondents from getting earlier treatment for severe mental health issues. The negative evaluations of professionals derived from negative past experiences and mistrust of the mental health professionals possibly deter a person from seeking help (125).” (Line 426-433)

g) Line 469-470. Think about re-writing to avoid repetitions

- Response: Thank you for this response. Here is the improved version: 

“The final analysis revealed that those with higher self-stigma scores were likely to have anxiety symptoms, whereas lower mental health-seeking attitudes predicted higher depressive symptoms. Poor help-seeking attitude prevents the low-income respondents from getting earlier treatment for severe mental health issues. The negative evaluations of professionals derived from negative past experiences and mistrust of the mental health professionals possibly deter a person from seeking help (125).”(Line 426-433)

h) Line 477. Delete 'a’ in ‘…of health a promotion’

- Response: “a” was removed at line 439 

i) Line 494. Did you mean ‘…structural issues were more likely to experience…’

- Response: Thank you for your response. The correction has been made at line 453

j) Line 538-539. One third had low resilience and higher resilience scores. Isn't is mutually exclusive? 

- Response: Apologies for the confusion. The sentence has now been rephrased to: 

“Approximately one-third of respondents had low resilience. Higher resilience scores associated with lower depressive and anxiety symptoms.”(line 470-471)

k) Line 539-540. ‘Substantial empirical studies have supported the inverse resilience scores concerning depression and anxiety’. Not sure what this sentence means. 

- Response: Apologies for the confusion. The sentence now rephrased to: 

“The findings were supported by a meta-analysis revealing of the association between inversed resilience scores with depression and anxiety (135).”(line 471-473)

Reviewer # xx

1. Abstract: 

a) Clearly written and concise.

- Response : Thank you

2. Introduction

a) Presented relevantly.

- Response : Thank you

3. Methods

a) Clearly presented.

b) However, regarding two of the tools used: 

1. EQ-5D

2. SCSRFQ

As mentioned in the manuscript (line 134: The respondents completed multiple standardized instruments in the Malay language with assistance from the research team), were the tools mentioned above translated and validated in Malay language for this study? The other tools were clearly informed with reference regarding its validity of Malay language questionnaires. 

- Response : 

Both tools were validated in the Malay language but among different populations (1. Shafie et al & 2. Mohamad et al ). Therefore they were validated in the initial pilot test and results showed a Cronbach alpha score of more than 0.80. 

The statement has been amended to :

“The Malay language of SCSRF has a Cronbach’s alpha of 0.84 and was validated in the current research’s pilot study with a good internal reliability of Cronbach’s alpha of 0.79.”(line 174-175)

1. Shafie A, Vasan Thakumar A, Lim C, Luo N. Psychometric performance assessment of Malay and Malaysian English version of EQ-5D-5L in the Malaysian population. Qual Life Res. 2018.

2. Mohamad AS, Draman S, Aris MAM, Musa R, Rus RM, Malik M. Depression, anxiety, and stress among adolescents in Kuantan and its association with religiosity-a pilot study. IIUM Medical Journal Malaysia. 2018;17(2).

4. Statistical Analysis:

a) Not an expert to comment on this but the presentation of data was clear and easily understood. 

- Response : Thank you

5. Results: 

a) No comment

- Response : Thank you

6. Discussion:

a) It addresses the results adequately and well thought through. Findings are highlighted with relevant suggestions on the importance of mental health and psychosocial interventions that can be taken up to policymakers. Limitations were addressed. 

- Response : Thank you

---

## [Editor Report · Decision Letter 2]

21 Jul 2022

Psychosocial factors associated with mental health and quality of life during the COVID-19 pandemic among low-income urban dwellers in Peninsular Malaysia

PONE-D-22-05096R2

Dear,

We’re pleased to inform you that your manuscript has been judged scientifically suitable for publication and will be formally accepted for publication once it meets all outstanding technical requirements.

Kind regards,

Muhammad Shahzad Aslam, Ph.D.,M.Phil., Pharm-D

Academic Editor

PLOS ONE
---

## [Editor Report · Acceptance letter]

4 Aug 2022

PONE-D-22-05096R2 

Psychosocial factors associated with mental health and quality of life during the COVID-19 pandemic among low-income urban dwellers in Peninsular Malaysia 

Dear Dr. Said:

I'm pleased to inform you that your manuscript has been deemed suitable for publication in PLOS ONE. Congratulations! Your manuscript is now with our production department. 

Kind regards, 

on behalf of

Dr. Muhammad Shahzad Aslam 

Academic Editor

PLOS ONE